# Birth order dependent growth cone segregation determines synaptic layer identity in the *Drosophila* visual system

Abhishek Kulkarni[1], Deniz Ertekin[1], Chi-Hon Lee[2], Thomas Hummel[1]*

[1]Department of Neurobiology, University of Vienna, Vienna, Austria; [2]Section on Neuronal Connectivity, Laboratory of Gene Regulation and Development, Eunice Kennedy Shriver National Institute of Child Health and Human Development, Bethesda, United States

**Abstract** The precise recognition of appropriate synaptic partner neurons is a critical step during neural circuit assembly. However, little is known about the developmental context in which recognition specificity is important to establish synaptic contacts. We show that in the *Drosophila* visual system, sequential segregation of photoreceptor afferents, reflecting their birth order, lead to differential positioning of their growth cones in the early target region. By combining loss- and gain-of-function analyses we demonstrate that relative differences in the expression of the transcription factor Sequoia regulate R cell growth cone segregation. This initial growth cone positioning is consolidated via cell-adhesion molecule Capricious in R8 axons. Further, we show that the initial growth cone positioning determines synaptic layer selection through proximity-based axon-target interactions. Taken together, we demonstrate that birth order dependent pre-patterning of afferent growth cones is an essential pre-requisite for the identification of synaptic partner neurons during visual map formation in *Drosophila*.

*For correspondence: thomas. hummel@univie.ac.at

**Competing interests:** The authors declare that no competing interests exist.

## Introduction

The identification of mechanisms that regulate the precise formation of neural circuits has been one of the major goals in developmental neurobiology. The Chemoaffinity hypothesis, formalized by Roger Sperry (*Sperry, 1963*), suggests that growing neurons must carry individual identification tags that allow the recognition between synaptic partners. Although many cell-type specific recognition molecules essential for neural circuit assembly have been identified in recent years (reviewed in *Missaire and Hindges, 2015*; *Yogev and Shen, 2014*), the precise developmental context in which these molecular tags control cell recognition and specify synaptic identity remain largely elusive.

The temporal pattern in which different types of neurons are generated and specified has been shown to influence their connectivity during further development (*Kohwi and Doe, 2013*; *Osterhout et al., 2014*; *Pujol-Marti et al., 2012*). In addition to cell type specification, all neurons undergo similar steps of cellular differentiation including the growth of specific processes with different molecular and functional properties (*Rolls, 2011*; *Tahirovic and Bradke, 2009*) accompanied by the expression of general neuronal molecules like N-Cadherin (*Gärtner et al., 2015*). These molecules common to most neurons also influence axon targeting (*Brusés, 2011*; *Sakai et al., 2012*) and synaptogenesis (*Basu et al., 2015*; *Bekirov et al., 2008*; *Seong et al., 2015*) but how their ubiquitous expression can support neuronal recognition is not well understood.

The *Drosophila* visual system, due to its highly stereotypic arrangement and genetic tractability, provides an excellent system to understand the mechanisms involved in neural circuit assembly (*Clandinin and Zipursky, 2002*). Each of the compound eyes is composed of approximately 800

**eLife digest** A nervous system requires a precise network of connections between cells called neurons to work properly. Within the brain, the fiber-like connections between pairs of neurons are not running crisscross like a pile of spaghetti. Instead, connected partner neurons are organized into distinct layers and columns.

Many questions remain about how these partner neurons find each other and how the layers of fiber-like connections form. To answer these questions, scientists often study the part of the fruit fly nervous system that controls the insect's vision. This brain-like structure is simple and can be easily manipulated with genetic engineering. Fruit fly studies have helped identify some molecules that play a role in helping partner cells find one another and connect. These studies have also shown that the timing of brain cell development appears to play a role. But the role that layer formation plays in the process is still a mystery.

Now, Kulkarni et al. show that the birthdate of neurons in the fruit fly visual system helps organize them into layers. These neurons are generated early in the development of the fly. Shortly after birth, a molecular clock under the control of a protein called Sequoia starts within each newly generated neuron. The Sequoia protein is a transcription factor and controls the activity of many genes, and the molecular clock provides the growing neuron fibers with information about where and when to look for its partner neurons.

By manipulating the amount and time that Sequoia is produced in the fly visual system, Kulkarni et al. show that this clock helps arrange the growing cells into layers. Cells with similar birthdates connect and are arranged into layers. How much and when Sequoia is produced dictates where each new layer begins. The next steps for the research will be to learn more about how the clock works and identify any intermediaries between the clock and cell growth patterns.

units called ommatidia (*Campos-Ortega, 1980*) and each ommatidium contains eight photoreceptor or retinula cells (R1-R8). Axons of R1-R6 photoreceptors terminate in the outermost lamina neuropile (*Fischbach and Dittrich, 1989*). In contrast, R8 and R7 axons project topographically through the lamina and terminate in the medulla (*Figure 1 A*). This topographic projection leads to the formation of medulla columns that receive input from R7/R8 cells of the same ommatidium. Within the medulla column R8/R7 axons terminate in two different layers, M3 and M6 respectively (*Fischbach and Dittrich, 1989*), in which they contact their post-synaptic partner neurons (*Fischbach and Dittrich, 1989*; *Gao et al., 2008*; *Karuppudurai et al., 2014*; *Melnattur and Lee, 2011*; *Ting et al., 2014*).

The layer specific targeting of R7/R8 axons can be divided into two main developmental phases: First, targeting of R cell axons to distinct temporary layers in the early medulla and second, the selection of correct synaptic target layer within the mature medulla neuropile (*Hadjieconomou et al., 2011*; *Ting and Lee, 2007*; *Ting et al., 2005*). Multiple cell type specific molecules involved in layer specific targeting of R8 axons (*Hakeda-Suzuki and Suzuki, 2014*; *Lee et al., 2003*; *Ohler et al., 2011*; *Pappu et al., 2011*; *Senti et al., 2003 Shinza-Kameda et al., 2006*; *Timofeev et al., 2012*; *Tomasi et al., 2008*) and R7 axons (*Astigarraga et al., 2010a*; *2010b*; *Choe et al., 2006*; *Lee et al., 2001*; *Morey et al., 2008*; *Nern et al., 2005*; *Prakash et al., 2010*; *Ting et al., 2005*) have been identified. Interestingly, most of these molecules function in the second phase of axon targeting and the molecular mechanisms governing the initial innervation of R8 and R7 axons as well as the importance of this temporary positioning for subsequent synaptic layer targeting remain elusive. The Zn finger transcription factor Sequoia and the cell adhesion molecule N-Cadherin are both expressed in R7 as well as R8 cells but are primarily required for the temporary layer positioning of R7 axons (*Lee et al., 2001*; *Petrovic and Hummel, 2008*; *Ting et al., 2005*). Additionally, the LRR molecule Capricious, expressed only in R8 cells, has been described to control R8 axon targeting during the second step (*Shinza-Kameda et al., 2006*) but its role in the initial targeting of R8 axons to temporary medulla position has not been addressed despite an early onset of expression.

Here we show that early self-patterning of R7/R8 afferents, mediated by relative difference in Sequoia levels, organizes an initial topographic map. This is achieved by a birth-order defined

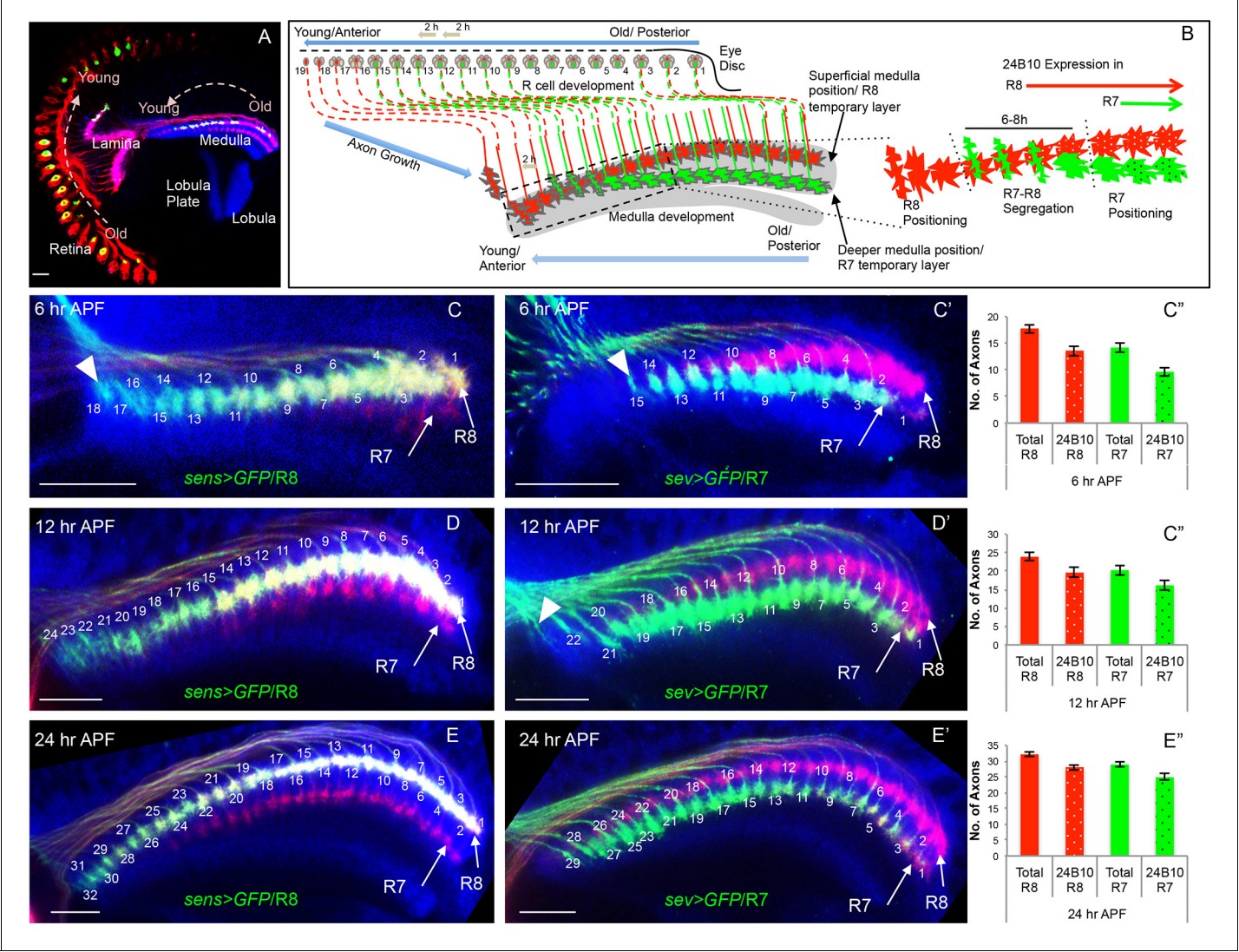

**Figure 1.** Initial positioning of R cell growth cones in the developing medulla target field. (A) Overview of the developing *Drosophila* visual system at 24 hr APF. Arrows indicate the developmental gradient of photoreceptor differentiation in the retina and corresponding axonal targeting in the medulla neuropile. (B) Model of initial innervation of R cell axons and growth cone segregation in the medulla. C–E''. R7/R8 axon innervation in the medulla target field at different developmental stages. (C, D, E) R8 growth cones labelled with *UAS-mCD8-GFP* expressed under *sens-Gal4*. C', D', E'. R7 growth cones labelled with *UAS-mCD8-GFP* expressed under *sev-Gal4*. C, C'. At 6 hr APF, 18 R8 (C arrowhead) and 15 R7 (C' arrowhead) axons innervate the medulla. D, D'. At 12 hr APF, 24 R8 and 21 R7 axons innervate the medulla. In addition to 21 R7 axons that are already present in the medulla field, 22nd R7 axon can be seen entering at the anterior medulla (D' arrowhead). E, E'. At 24 hr APF, 32 R8 and 29 R7 axons have innervated the medulla field. Scale bar shown in all images is 20μm. All photoreceptor axons are visualized using 24B10 antibody (*Fujita et al., 1982* in red), R8 (in A, C, D, E) and R7 growth cones in (C', D', E') are stained with anti-GFP antibody (in green) and medulla neuropile is stained using anti-N-Cadherin antibody (in Blue). C'', D'', E''. Quantification of the sequential innervation of R7/R8 axons in the medulla field. Error bars indicate Standard Deviation.

The following source data and figure supplement are available for figure 1:

**Source data 1.** R cell innervation quantification.
**Source data 2.** Sequoia expression quantification.
**Figure supplement 1.** Onset of cell-type specific marker expression in the developing R cells.

sequence of R7-R8 growth cone segregation leading to their differential positioning in the target field. Shortly afterwards, cell adhesion molecules, like Capricious in R8, consolidate these growth cone positions, which is critical for subsequent steps of synaptic partner recognition.

## Results

### R-cell axon innervation in the medulla mirrors their temporal pattern of specification

In the developing *Drosophila* visual system, photoreceptor axons project from the eye disc into the optic lobe and target to the lamina and medulla neuropiles (*Figure 1A*). Photoreceptor differentiation begins in the 3rd instar eye disc in a defined sequential fashion with R8 specified first followed by the outer R1-R6 cells and finally R7 in every ommatidium and can be visualized in developing ommatidial rows (*Figure 1B*, *Tomlinson and Ready, 1987*).

We examined how R8/R7 sequential specification is represented in the arrival of R cell axons at the medulla target region using cell type specific reporter lines (*sens-Gal4* for R8, *sev-Gal4* for R7, *Figure 1C–E"*, *Figure 1—figure supplement 1*). By quantifying the number of R8/R7 growth cones at three consecutive stages of early pupal development (6/12/24 hr After Puparium Formation, APF), we could show that R cell axon innervation in the medulla mirrors the temporal pattern of R cell specification and follows a consistent sequence of growth cone segregation for each stage (*Figure 1C–E'*, summarized in B). At the anterior edge of the medulla the youngest R8 axons arrive sequentially and position at the superficial layer of the medulla neuropile ('R8 positioning', *Figure 1C* arrowhead), followed by the arrival of the first R7 axon at the respective R8 position about 6 hr later and three columns posterior to the youngest R8 axon (*Figure 1C'* arrowhead). For the next 6-8 hr, represented by 3–4 columns, R7/R8 growth cones are in close contact followed by their segregation into adjacent positions, with R7 growth cones locating proximal with respect to the superficial R8 growth cones in the corresponding columns ('R7-R8 segregation', *Figure 1C'-E'*). Afterwards, R7 growth cones move deeper into the medulla neuropile whilst maintaining their columnar topography and a separate R7 temporary layer becomes visible ('R7 positioning', illustrated in *Figure 1B*).

### Sequoia mediates columnar as well as layer segregation of R7 and R8 growth cones in the medulla

As reported previously, Sequoia is critical for R cell target layer selection (*Petrovic and Hummel, 2008*). Next we tested if Sequoia is involved in the regulation of these initial steps of growth cone positioning. Using MARCM (*Lee and Luo, 2001*) we generated *sequoia* mutant R8 cells by activating Flippase under heat shock promoter and visualized their growth cones with an R8 specific reporter line (*sens-Gal4*, *Figure 2A–B'*). Axons of *sequoia* mutant R8 cells project to the anterior medulla in the wild type sequence but convergence of neighbouring R8 growth cones can be observed, disrupting the topographic organization (*Figure 2A–B'*). Similar phenotype in R8 growth cone positioning can be observed in eye specific *sequoia* mosaics (ey3.5-FLP), excluding an effect of unlabelled sequoia mutant cells generated in the brain in the hs-FLP background (data not shown).

Similar to R8, no defects can be detected in the extension and arrival of *sequoia* mutant R7 axons but they fail to segregate from R8 growth cones within the same column as early as 6 hr APF (*Figure 2C–D'*). Subsequently, these *sequoia* mutant R7 growth cones remain at the superficial medulla position together with R8 growth cones even at 24 hr APF (*Figure 2E–F'*). Interestingly, *sequoia* mutant R7 growth cones from two neighbouring columns also converge in the superficial medulla position (*Figure 2G–G'*). Single cell analysis revealed a cell-autonomous function of Sequoia in growth cone segregation (*Figure 2—figure supplement 1A–B'*, arrowhead). Therefore, loss of Sequoia function disrupts the sequential segregation of R8-R8, R8-R7 and R7-R7 growth cones within as well as between layers.

The targeting phenotype of *sequoia* mutant R7 cells varies depending on the genetic background of R8 in the same medulla column: In case of wild type R8 axons, *sequoia* mutant R7 from the same ommatidium will terminate together in M3 (*Figure 2I,I'*), whereas a sequoia mutant R8 and R7 co-terminate in M1 (*Petrovic and Hummel, 2008*). To test if mis-targeting of *sequoia* mutant R7 cell axons is caused by a change in their target layer recognition or the consequence of an afferent segregation defect, we analysed the phenotype of wild type and *sequoia* mutant R7 axons in a

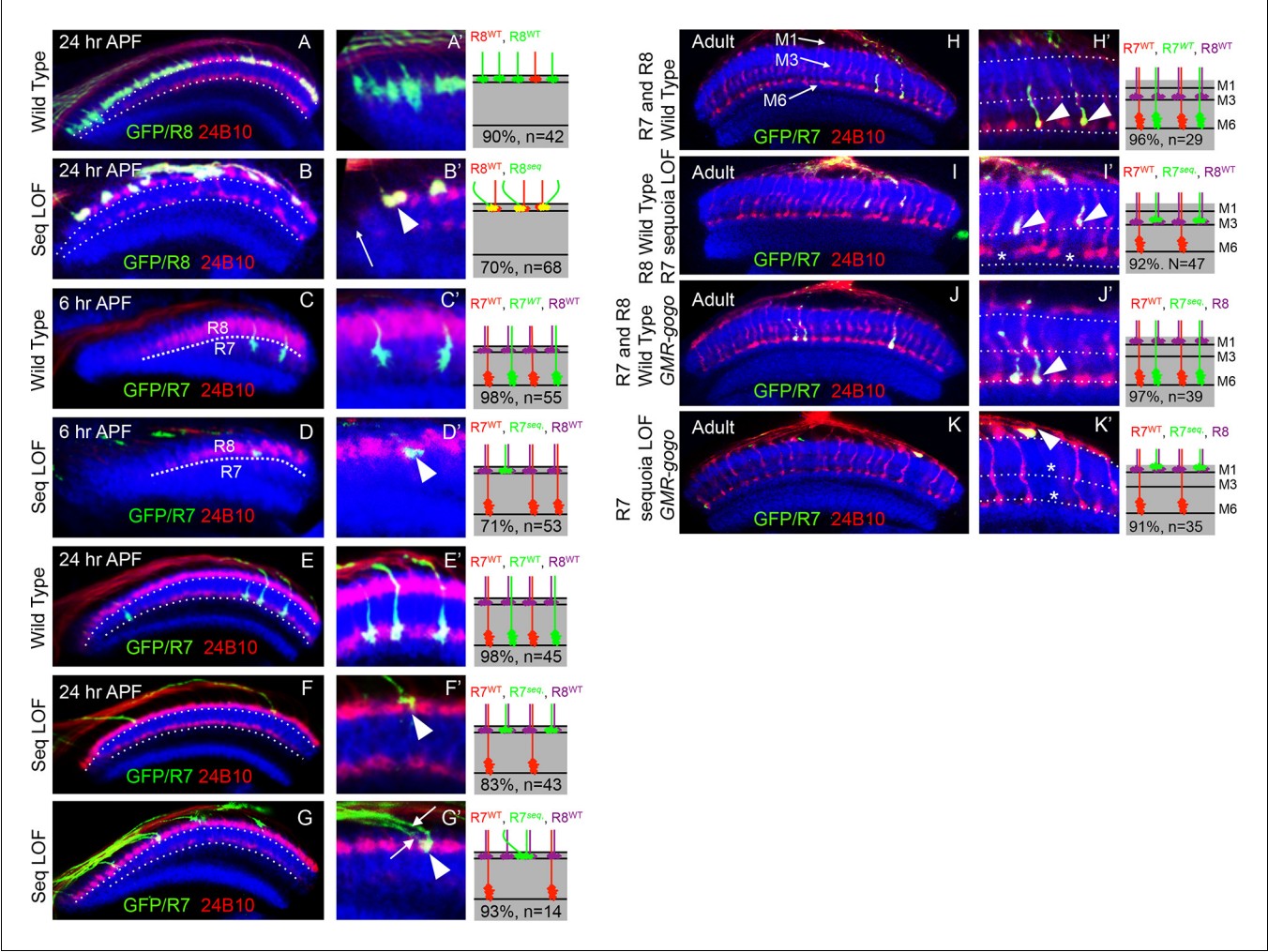

**Figure 2.** Sequoia mediates growth cone segregation of R cell axons in the medulla. (A, A') Wild type position of R8 growth cones as they arrive at the anterior region of the medulla (A'). (B, B') *sequoia* mutant R8 growth cones converge upon arrival at the anterior medulla (B' arrowhead) leaving gaps in their normal position (B' arrow). (C, C') Growth cones of wild type R7 cells segregate from R8 growth cones at 6 hr APF and are positioned immediately proximal to the R8 growth cones in respective columns. (D, D') Growth cones of *sequoia* mutant R7 cells fail to segregate from R8 growth cones and are positioned with R8 growth cones at the superficial medulla position at 6 hr APF (D' arrowhead). (E, E') Wild type R7 growth cones reach their temporary target layer in the deeper medulla position at 24 hr APF. F, F'. *sequoia* mutant R7 growth cones fail to reach their temporary target layer and remain at the superficial medulla position with R8 growth cones at 24 hr APF (G' arrowheads). (G, G') Two neighbouring *sequoia* mutant R7 growth cones (G' arrows) converge into a single column (G' arrowhead) in the superficial medulla position. (H, H') Wild type R7 axons target to medulla layer M6 and R8 axons target to layer M3 in the adults (H' arrowhead). (I, I') *sequoia* mutant R7 axons mis-target to M3, the target layer for R8 axons (I' arrowhead) leaving layer M6 empty (I' asterisk). (J, J') Wild type R7 axons target to medulla layer M6 in the adult (J' arrowhead) even when R8 axons are retained in layer M1 due to expression of Golden Goal (*GMR-gogo*). (K, K') *sequoia* mutant R7 axons in presence of *GMR-gogo* mis-target to medulla layer M1 along with R8 axons (K' arrowheads) leaving both layers M3 and M6 empty (K' asterisks). Schematics in all panels illustrate growth cone positioning (A–G') or axon targeting (H–K') phenotypes and numbers indicate quantification of respective phenotype.

The following figure supplement is available for figure 2:

**Figure supplement 1.** Cell-autonomous effects of Sequoia loss-of-function in R8 cells (A, A') Wild type single R8 cell clones.

background with a modified R8 axon position. R8 axons were retained in the superficial medulla layer M1 via Golden-goal (*gogo*) over-expression using *GMR-gogo*, which has no effect on the targeting of R7 axons (*Tomasi et al., 2008*). In this background, we labelled wild type and *sequoia* mutant R7 cells using MARCM and analysed axon targeting in the adult medulla. Wild type R7 growth cones segregate from ectopic R8 growth cones and reach the layer M6 (*Figure 2J–J'*), indicating that the ectopic position of R8 axons does not influence wild type R7 axon targeting. In

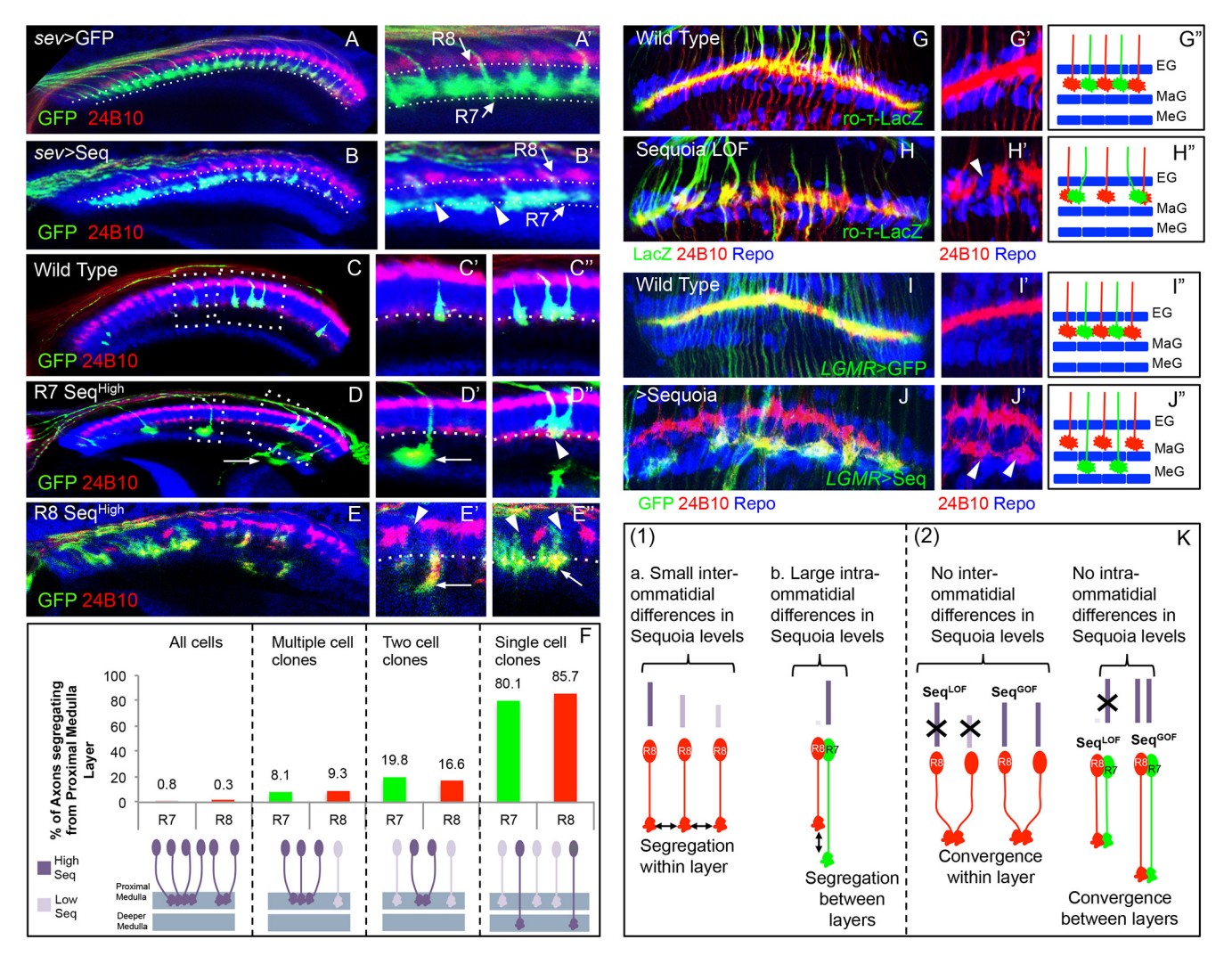

**Figure 3.** Relative levels of Sequoia mediate growth cone segregation. (**A**) Wild type R7 growth cones segregate within the deeper medulla position thereby maintaining the topographic columnar arrangement. (**B**) Mis-expression of Sequoia in R7 cells alone under *sev-Gal4* (all R8 Seq[low], all R7 Seq[high]) does not affect segregation of R7 growth cones from R8 growth cones but disrupts segregation among R7 growth cones within deeper medulla position. This leads to the loss of topographic arrangement illustrated by gaps in the deeper medulla position (**B'** arrowheads). R8 and R7 growth cone positions in the medulla are indicated in **A'–B'**. **C–E"**. Wild type and Sequoia mis-expressing clones of R7 cells are generated using *GMR-FLP* induced MARCM. (**C'**) shows a single cell and **C''** shows a two cell R7 clone (**C', C''** arrowheads). (**D, E**) Sequoia mis-expressing clones of R7 and R8 respectively. **D'** shows a single Sequoia mis-expressing R7 cell clone (R7 Seq[high]) that segregates from growth cones of neighbour wild type R7 cells (R7 Seq[low]) and extends beyond the normal R7 position into the medulla; **D''** shows a two cell clone of neighbouring R7 cells that mis-express Sequoia (R7 Seq[high]-R7 Seq[high]) but do not extend growth cones beyond the normal R7 position in the medulla (**D''** arrowhead). (**E'**) Single Sequoia mis-expressing R8 cell clone (R8 Seq[high]) extends growth cone to the medulla regions beyond the R7 position (**E'** arrow) thus leaving a gap in the superficial R8 position in the medulla (**E'** arrowhead). (**E"**) Multiple R8 cell clones exhibit shift of their growth cones to deeper medulla position (**E"** arrowheads showing empty R8 position) but are retained in this position similar to the two cell R7 clones (R8 Seq[high]-R8 Seq[high] **E"** arrow). The brain regions are visualized using anti-N-cadherin antibody (in blue) and labelling of photoreceptors axons is indicated in the figure panels. (**F**) Quantification of the overshooting phenotype at the Seq[low]-Seq[high] clone boundary exhibited by single vs. multiple cell clones of R7 and R8 cells. Sequoia expression in- all R cells- R7 n=96, R8 n=128, multiple cell clones- R7 n=37, R8 n=43, two cell clones- R7 n=97, R8=6 and single cell clones- R7 n=121, R8 n=7. (**G–J"**) Lamina plexus assembly is disrupted by loss and gain of Sequoia function. (**G, G"**)- Wild type clones of lamina targeting R1-R6 cells are generated using *ey3.5-FLP* and are labelled with *ro-τ-LacZ* to visualize growth cones of R2/R5 cells. (**H, H"**) Disruption of lamina plexus assembly in *sequoia* mutant R1-R6 clones (**H'** arrowhead). R2/R5 cell growth cones visualized with *ro-τ-LacZ*. (**I, I"**) Wild type clones of R1-R6, generated using *ey3.5-FLP* and labelled with *LGMR-Gal4*, target normally to lamina plexus. (**J, J"**) Sequoia mis-expressing clones of R1-R6 cells labelled with *LGMR-Gal4*. Growth cones of Sequoia mis-expressing R1-R6 cells (labelled with GFP) segregate from the growth cones of wild type cells and assemble into an additional layer between marginal (MaG) and medulla glia (MeG) cells (**J'** arrowheads). **G"–J"** show schematics of wild type, *sequoia* mutant and Sequoia mis-expressing R1-R6 growth

*Figure 3 continued on next page*

*Figure 3 continued*
cones in the lamina plexus. (**K**) Schematics depicting (1) the relative differences in Sequoia expression levels among R8-R8 and R8-R7 cells in wild type development, (2) R cell growth cone phenotypes in Sequoia loss and gain of function scenarios.
The following source data and figure supplement are available for figure 3:

**Source data 1.** R cell axon overshooting quantification.
**Figure supplement 1.** Loss and gain of Sequoia function in single lamina targeting R cell-type (R4) is sufficient to induce changes in growth cone segregation.

contrast, *sequoia* mutant R7 axons mis-target to layer M1 along with ectopic R8 axons (*Figure 2K–K'*). This indicates that mis-targeting of sequoia mutant R7 axons is the consequence of growth cone segregation defects rather than a change in target layer recognition.

In summary, Sequoia regulates two main steps of growth cone segregation of R7/R8 axons. For axons of the same R cell type but from different ommatidia, Sequoia supports point-to-point spacing within the temporary layer and thereby controls the subsequent columnar restriction. In addition, for the pair of R7/R8 cells from a single ommatidium, which innervate the same medulla column, Sequoia mediates the segregation of growth cones between layers.

## Relative difference in Sequoia expression determine R cell growth cone segregation

Sequoia shows a highly restricted expression during early differentiation of all R cells (*Petrovic and Hummel, 2008*). A short peak of high Sequoia expression at the onset of R cell differentiation is followed by a rapid decline in protein levels (*Figure 1—figure supplement 1A* graph). Due to the sequential development of ommatidia in the eye field as well as R cell types within each ommatidium (*Tomlinson and Ready, 1987*), the short peak of Sequoia expression in each cell leads to small differences of Sequoia levels in sequentially projecting R8 cells from adjacent ommatidial rows ('*inter-ommatidial differences*', *Figure 1—figure supplement 1A*, *Figure 3K1a*). In addition, each R7 develops approximately 8 hr after R8 in every ommatidium corresponding to four ommatidial rows in the eye disc. Therefore, the highest difference in the expression levels of Sequoia can be found between these two cell types with almost no detectable Sequoia in R8 at the time of maximal Sequoia expression in the R7 of the same ommatidium ('*intra-ommatidial differences*', *Figure 3—figure supplement 1C*, inset 3, *Figure 3K1b*).

To determine the role of Sequoia expression dynamics in R7/R8 growth cone positioning we next analysed how inter- and intra-ommatidial differences in Sequoia levels influence growth cone segregation. Abolishing the inter-ommatidial differences via prolonged Sequoia expression using *LGMR-Gal4* leads to a convergence of R7/R8 growth cones in the deeper medulla position (*Figure 3—figure supplement 1A–B'*). Similarly, equalizing Sequoia levels in only R7 cells using *sev-Gal4,* which shows transient, low-level expression as compared to *LGMR-Gal4*, resulted in frequent R7 growth cone convergence and corresponding innervation gaps in the deeper medulla layer (*Figure 3A–B'*, arrowheads in B'). This result shows that the loss and gain of Sequoia activity leads to similar growth cone convergence phenotypes indicating that the presence of Sequoia expression is not sufficient to mediate growth cone segregation.

To determine if the observed differences in Sequoia levels between R cells are critical for their growth cone segregation into different layers, we sought to model the endogenous Sequoia expression difference between R8 and R7 cells among neighbouring R7 cells targeting within a layer. For this, we analysed the axon projections in Sequoia gain-of-function R7 cell mosaics (Seq^high) at the clonal boundary with adjacent wild type R7 cells (Seq^low; *Figure 3C–F*). Here we observed that the majority of single Seq^high R7 axons segregate from the surrounding wild type Seq^low axons and project deeper into the medulla (*Figure 3D'*). In contrast to the single cell clones, most of the mosaics with two adjacent Seq^high R7 cells show growth cone convergence and termination at the appropriate medulla position (*Figure 3D''*). Interestingly, single Seq^high R7 cells from distant columns, which overshoot the temporary layer, often converge in the inner medulla neuropile (*Figure 3D'* arrow) or even within the lobula complex (data not shown). Similar to R7, mosaics of R8 cells with different Sequoia expression

levels segregate during initial axon positioning: single Seq[high] R8 axons segregate from the surrounding wild type Seq[low] R8 axons (*Figure 3E'* arrowhead) and extend further into the medulla neuropile (*Figure 3E'* arrow), whereas two or more neighbouring Seq[high] R8 growth cones converge and terminate at the R7 position (*Figure 3E''* arrow). Thus, together with the loss-of-function, this gain-of-function analyses show that 'equalized' Sequoia levels prevent growth cone segregation thereby highlighting the role of relative Sequoia levels in this process (summarized in *Figure 3K2*).

To test if relative differences in Sequoia levels also control growth cone patterning of other photoreceptors, we analysed the outer R1-R6 cells. These neurons develop in a temporal window between R8 and R7 specification (*Tomlinson and Ready, 1987*) and their growth cones are positioned in a single temporary layer, the lamina plexus, which corresponds to their small relative differences in Sequoia expression (*Figure 3—figure supplement 1C*, *Figure 3G,G'*), We generated *sequoia* mutant clones of R1-R6 cells using *ey3.5-FLP* and visualized the growth cones of R2/R5 cells (*ro-τ-LacZ*) or R4 cells (*mδ0.5-Gal4*) (see Materials and methods). The *sequoia* mutant R1-6 growth cones fail to segregate normally, illustrated by gaps within the lamina plexus (*Figure 3H–H''*, *Figure 3—figure supplement 1E-E'*), which is similar to the defects in inter-ommatidial R8 as well as R7 growth cone segregation. Strikingly, clones of Seq[high] R1-6 cells segregate their growth cones from the wild type (Seq[low]) ones thereby organizing the formation of a novel layer within the lamina neuropile (*Figure 3I–J''*, *Figure 3—figure supplement 1F,F'*). This demonstrates that large differences in Sequoia levels among R cells, either inherently present as in case of developing R7/R8 cells or ectopically generated as in case of R1-R6 cells, segregate their growth cones into distinct layers. In all cases, cells with higher Sequoia levels position their growth cones deeper in the neuropile. From these data we propose a 'Growth Cone Segregation' model, in which small differences in Sequoia levels lead to an evenly–spaced growth cone positioning within a layer, whereas large Sequoia differences result in segregation of growth cones into separate layers. In contrast, equal levels of Sequoia in projecting R cells, either in loss- or gain-of-function context, cause the convergence of growth cones and termination of axonal extension (*Figure 3K*).

## Initial position of R8 growth cones in the medulla is developmentally consolidated

We next determined the temporal dynamics of initial R8/R7 growth cone segregation into superficial vs. deeper medulla layers. To temporally restrict the Gal4-driven expression of Sequoia, we utilized the Gal80[ts] (TARGET system, *McGuire, 2003*). At the permissive temperature (18°C) functional Gal80 prevents Sequoia expression whereas an inactive Gal80 at the restrictive temperature (29°C) allows the induction of Sequoia expression. Using this method, an R8 growth cone shift to deeper medulla position can be observed following Sequoia expression until 24 hr APF (*Figure 4—figure supplement 1C*). In contrast, an onset of Sequoia expression after 24 hr APF, at which time all photoreceptor axons have arrived in the target field, has no effect on R7/ R8 growth cone segregation (*Figure 4—figure supplement 1D*). This indicates that R7/R8 cells are sensitive to Sequoia-induced growth cone segregation only during a narrow developmental window. Interestingly, although R8 growth cones do not change the layer upon Sequoia expression after 24 hr APF, they still respond to these elevated Sequoia levels by leaving their topographic position and converging onto neighbouring growth cones (*Figure 4—figure supplement 1D*). This indicates that the restriction of R cell growth cones to distinct temporary layers occurs immediately following their segregation whereas the process of columnar restriction continues into the later steps of visual map formation (*Ferguson et al., 2009*; *Ting et al., 2007*).

To further characterize the developmental sequence of the transition from initial positioning to the consolidation of R8 growth cones, we induced short pulses of Sequoia expression in early pupal stages using the Gal80[ts] method described above and analysed R8 growth cones at 24 hr APF (*Figure 4A–F*). Upon the induction of Sequoia expression at 6 hr APF with about 18 R8 growth cones in the medulla target region, we observed a defined shift of the youngest 10 of these R8 growth cones to the deeper R7 position whereas the more posterior, and therefore older, 8 R8 growth cones remain at their superficial medulla position. This indicates that posterior R8 growth cones had consolidated their position prior to the effects of induced Sequoia expression (*Figure 4B*). Two hours later (at 8 hr APF), four more R8 growth cones (12 out of 20) are no longer responsive to elevated Sequoia levels as they do not leave their superficial medulla positions (*Figure 4C*). Upon the onset of Sequoia expression at 12 hr APF, 16 out of 24 R8 growth cones are retained in the

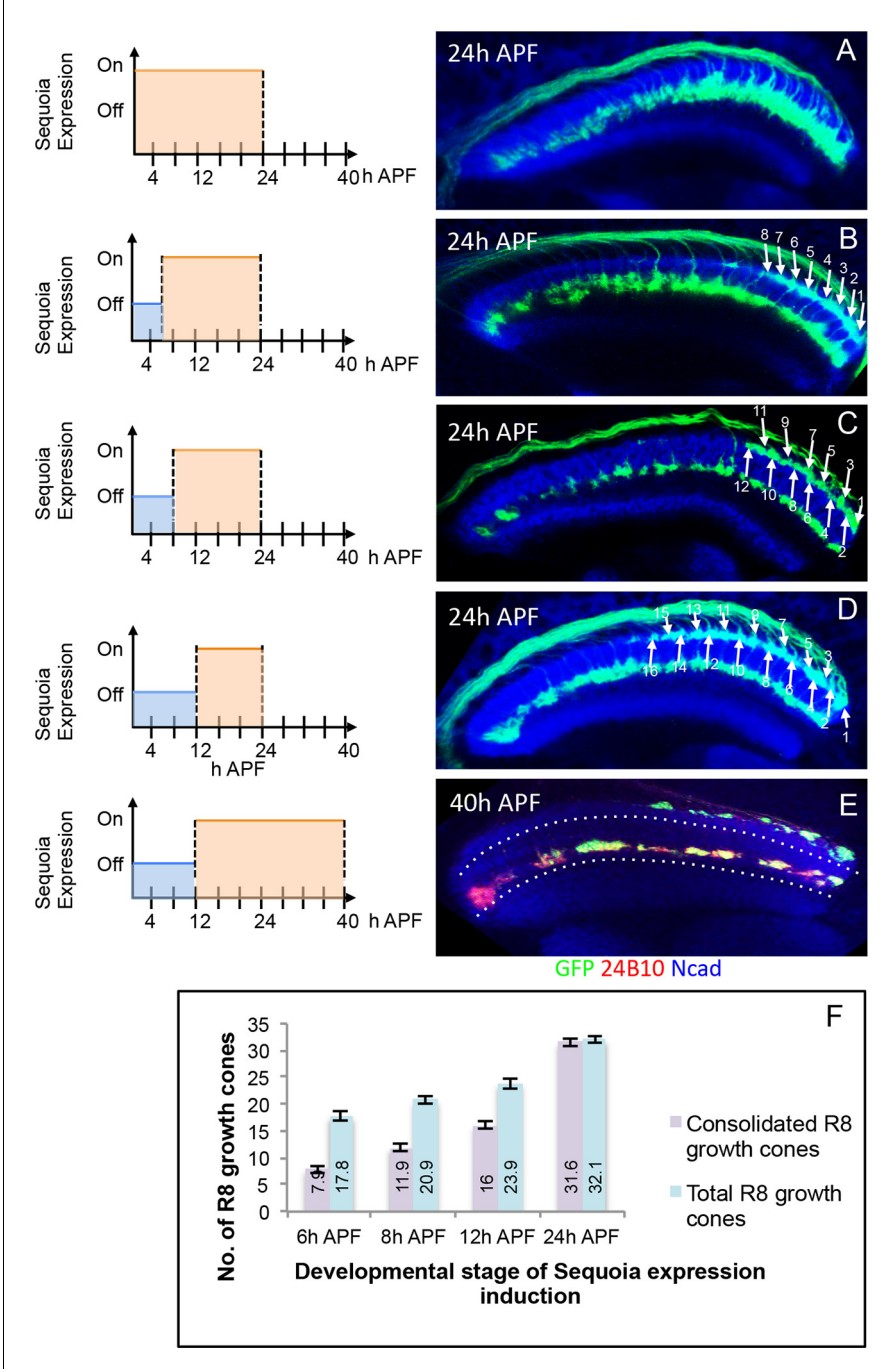

**Figure 4.** Initial position of R cell axons in the medulla is developmentally consolidated. (**A**) Early mis-expression of Sequoia from 3rd instar stage leads to shift of all R8 growth cones to deeper medulla position and results in convergence with R7 growth cones. (**B**) Induction of Sequoia mis-expression from 6 hr APF shows consolidation of 8 posterior R8 growth cones in their superficial medulla position at 24 hr APF. (**C**) Sequoia mis-expression from 8 hr APF shows 12 posterior R8 growth cones to be consolidated. (**D**) Sequoia mis-expression from 12 hr APF onwards shows consolidation of 16 posterior R8 growth cones in superficial medulla position at 24 hr APF. (**E**) Continued Sequoia mis-expression until 40 hr APF following induction at 12 hr APF does not disrupt the consolidation of 16 posterior R8 growth cones. (**F**) Quantification of R8 growth cone consolidation. Error bars indicate Standard Deviation.

The following source data and figure supplement are available for figure 4:

*Figure 4 continued on next page*

*Figure 4 continued*

**Source data 1.** R8 axon consolidation quantification.
**Figure supplement 1.** Temporally restricted induction of Sequoia expression has differential effects on R7/R8 growth cone segregation.

superficial medulla position (*Figure 4D*). Additionally, continuous Sequoia expression at later developmental stages does not disrupt the superficial layer positioning of R8 growth cones that are consolidated prior to Sequoia induction (*Figure 4E*). Together, the consolidation of each R8 growth cone occurs approximately 18 hr after its arrival at the superficial medulla layer (*Figure 4F*). The fixed correlation between the number of consolidated R8 growth cones and developmental time indicates that for each R8 growth cone there is a defined transition from the initial positioning to the consolidation shortly after R7-R8 segregation.

## Capricious mediates the consolidation of R8 growth cones in the initial position

To gain further insights into the molecular mechanism underlying the consolidation of R8 cell growth cones we tested candidate molecules expressed during initial axon targeting. Capricious (Caps) is expressed in projecting R8 cells and has been proposed to mediate R8 axon targeting (*Shinza-Kameda et al., 2006*). We determined the relative levels of Capricious expression in the developing medulla at the R8 and R7 positions by measuring the ratio of normalized fluorescence intensities (NFI) of Capricious staining at the R8 and R7 growth cones against the surrounding medulla region (*Figure 5* Table). At the anterior medulla, Capricious shows a homogeneous expression with NFI ratios for R cell growth cones and corresponding medulla region being close to 1 (0.94 in R8 and 0.87 in R7 growth cone position, *Figure 5A–D*). Following the phase of growth cone segregation, Capricious enriches during the process of consolidation at the region of R8 growth cones with an NFI ratio increasing to 1.39 and a four fold decline at the R7 position (with the ratio of 0.24, *Figure 5A'–D'*). This indicates that during the process of consolidation, Capricious levels in R8 growth cones increase. At the same time, Capricious-negative R7 growth cones move to a deeper medulla position devoid of any Capricious expression.

To determine the role of Capricious in the process of growth cone consolidation we followed the projection of *capricious* mutant R8 cells using MARCM. Interestingly, no defect could be detected in the initial position of R8 growth cones as well as subsequent R7/R8 segregation (*Figure 5E–E",G,G'*). R8 growth cone positioning defects first appear during the phase of consolidation where *capricious* mutant growth cones extend towards the deeper R7 position (*Figure 5G"*). Later in development, these *capricious* mutant R8 axons mis-target to layer M6 as previously reported (*Figure 5F–H"*, *Shinza-Kameda et al., 2006*). This shows that Capricious is not required for the initial positioning of R8 growth cones and R7/R8 segregation but is critical for the subsequent step of R8 growth cone consolidation in the superficial medulla.

To further elucidate the role of Capricious in the consolidation of R8 growth cones we modified Capricious levels in R8 cells with prolonged Sequoia expression. The partial reduction of Capricious levels via targeted RNAi does not interfere with the consolidation of R8 growth cones in the superficial medulla position (*Figure 5I*). As described above, approximately 16 R8 growth cones in the posterior medulla region are consolidated in their initial position at 12 hr APF, making them insensitive to the induction of Sequoia expression (*Figure 5J*). In contrast, the RNAi-mediated Capricious reduction in the background of elevated Sequoia expression severely affected the R8 growth cone consolidation resulting in most of the posterior R8 growth cones shifting to the deeper R7 position (*Figure 5K,L*). Furthermore, co-expression of Sequoia and Capricious prevents Sequoia-induced shift of R8 growth cones and leads to segregation of R7 and R8 growth cones in a wild type pattern (*Figure 5—figure supplement 1A–B'*). Therefore, Sequoia mediated growth cone segregation and Capricious mediated adhesion are antagonistic forces that are capable of balancing each other to achieve proper positioning of R-cell growth cones in the developing medulla.

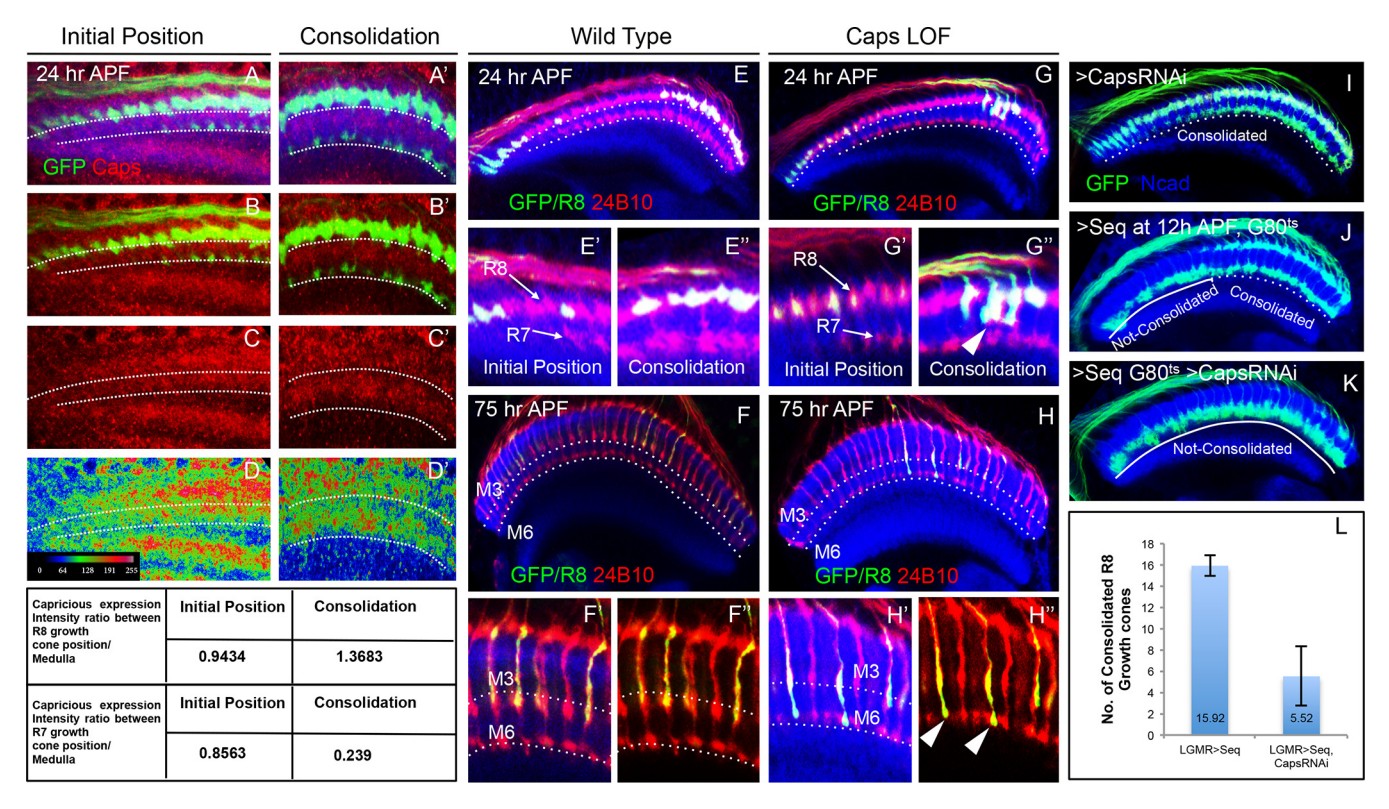

**Figure 5.** Capricious mediates initial position consolidation of R8 axons. (A–D') Expression pattern of Capricious protein in the developing medulla at 24 hr APF. A, B, C and D show Capricious expression at the anterior medulla corresponding to the initial positioning of R8/R7 growth cones as they innervate the medulla and A', B', C' and D' show the Capricious expression at the posterior side of medulla corresponding to the region where R8 growth cones are consolidated in their superficial medulla position. (D–D') The heat map of Capricious expression (measured in terms of normalized relative fluorescence intensity) in the developing medulla region. The Arbitrary Fluorescence Units used for plotting the heat map are shown in D. The table shows the quantification of Capricious expression intensity measured as ratio between the Normalized Fluorescence Intensities (NFI) at R8 or R7 growth cone position vs. the surrounding medulla region. (E–H") Loss of Capricious function disrupts R8 growth cone consolidation. (E, E") Wild type R8 clones at 24 hr APF. Wild type R8 growth cones are positioned at the superficial medulla position at both anterior (E') and posterior (E") side of medulla (98%, n=58). (F, F") Wild type R8 axons at 75 hr APF target to medulla layer M3 (F', F", 97%, n=49). (G–H") *capricious* mutant R8 clones. (G, G") *capricious* mutant R8 clones at 24 hr APF. *capricious* mutant R8 growth cones are positioned correctly at the anterior side of medulla (G', 94%, n=37) as they innervate medulla whereas at the posterior side (G") *capricious* mutant R8 growth cones prematurely extend towards the deeper medulla position (G" arrowheads, 62%, n=24). (H–H") *capricious* mutant R8 axons at 75 hr APF mis-target to medulla layer M6 instead of M3 (H" arrowheads, 71%, n=38). (I–L) Reduction of Capricious levels leads to the disruption of R8 growth cone consolidation. I. *UAS-Capricious^RNAi* expression under *LGMR-Gal4* does not affect R8 growth cone consolidation at 24 hr APF. (J) Sequoia mis-expression from 12 hr APF leads to consolidation of 16 posterior R8 growth cones at 24 hr APF. (K) Sequoia mis-expression from 12 hr APF in sensitized background of *UAS-Capricious^RNAi* severely disrupts R8 growth cone consolidation at 24 hr APF. (I, J, K) Dotted lines depict region in the medulla with R8 growth cone consolidation and solid lines depict region in the medulla with R8 growth cones extending to the deeper R7 position. (L) Quantification of the *UAS-Capricious^RNAi* mediated disruption of R8 growth cone consolidation. Error bars indicate Standard Deviation.

The following source data and figure supplement are available for figure 5:

**Source data 1.** UAS-Seq/ UAS-Seq; UAS-Caps RNAi consolidation quantification.

**Figure supplement 1.** Antagonistic interaction between Sequoia and Capricious mediates proper positioning of R-cell growth cones in the developing medulla.

## Initial R cell growth cone segregation determines final target layer selection

Following the sequential positioning of R7/R8 growth cones during the first half of pupal development, all R7 and R8 axons simultaneously extend during the second half of medulla circuit assembly

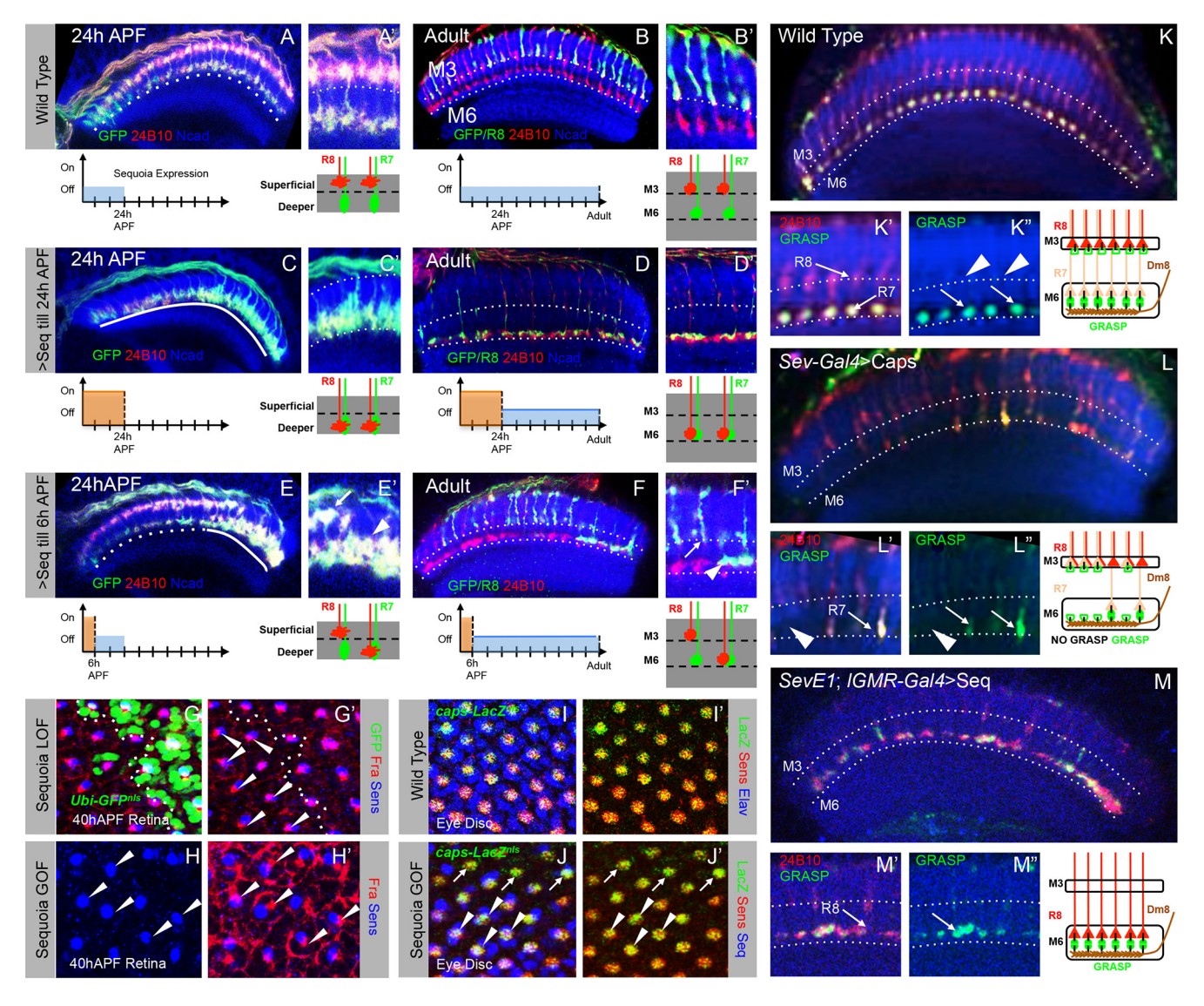

**Figure 6.** Initial position determines final medulla layer targeting and synaptogenesis. (**A–F'**) Initial growth cone position correlates with final target layer. (**A**) Wild type R7/R8 growth cone position in the medulla at 24 hr APF. (**B**) Wild type innervation of R8 axons in the adult to layer M3 and R7 axons to layer M6. (**C**) Induced Sequoia expression mediates mis-positioning of R8 growth cones in the deeper medulla position at 24 hr APF. (**D**) R8 axons mistarget to layer M6 in the adult even when Sequoia mis-expression is stopped from 24 hr APF onwards. (**E**) Sequoia mis-expression until 6 hr APF leads to mis-positioning of 6–8 posterior R8 growth cones to deeper medulla position. (**F**) Initially mis-positioned R8 growth cones mis-target to layer M6 whereas normally positioned R8 growth cones later target to layer M3. **A'-F'** shows magnifications of **A-F**. (**G–J'**) Changes in Sequoia expression do not affect expression of known R8 targeting molecules. (**G, G'**) Homozygous *sequoia* mutant cells are visualized using loss of *Ubi-GFP^{nls}* expression as a clonal marker. Arrowheads show individual R8 cells labelled with Senseless and Frazzled and cells without GFP are *sequoia* mutant R8 cells. (**H, H'**) Visualization of Frazzled expression following Sequoia mis-expression using *LGMR-Gal4*. All cells mis-express Sequoia and individual R8 cells are labelled with Senseless (Blue). Frazzled expression is visualized using anti-Frazzled antibody (Red). All R8 cells express Frazzled suggesting mis-expression of Sequoia does not repress Frazzled expression. Additional Frazzled staining at the ommatidial boundaries is from a different imaging plane. (**I–J'**) Elevated Sequoia levels do not affect Capricious expression. I, I'- Wild type pattern of Capricious expression reported by *caps-LacZ^{nls}* enhancer trap (Shishido et al., 1998) and visualized using anti-LacZ antibody (Green) in the R8 cells labelled with Senseless (Blue). (**J, J'**) Sequoia mis-expression does not transcriptionally repress the expression of Capricious (**J, J'**- Arrowheads show R8 nuclei with Sequoia mis-expression, arrows show R8 nuclei without Sequoia mis-expression). (**K–M"**) Synaptogenesis of R8 axons in the ectopic layer M6 with Dm8 neurites as shown by *GFP Reconstitution Across Synaptic Partners* (syb-GRASP). (**K–K"**) Control GRASP between R7 and Dm8 at layer M6. R8 and R7 axons target to layers M3 and M6 (**K'** arrows) and GRASP signal is observed in layer M6 (**K"** arrows) but not in layer M3 (**K"** arrowheads). (**L–L"**) Mistargeting of R7 axons to layer M3 upon *UAS-Capricious^{ID}* expression under *sev-Gal4* leads to loss of GRASP between R7 and Dm8. Escaper R7 axons that target to M6 show GRASP

*Figure 6 continued on next page*

*Figure 6 continued*

with Dm8 (L′ arrows) whereas R7 axons that mistarget to M3 do not show GRASP signal (L″ arrowheads). (M–M″) R8 axons ectopically targeting to layer M6 upon Sequoia mis-expression show GRASP signal with Dm8 (M″ arrows).

The following figure supplements are available for figure 6:

**Figure supplement 1.** Ectopic R8 growth cones are present in close proximity of Dm8 neurites early in the development.

**Figure supplement 2.** Role of N-Cadherin in stabilizing R7 growth cones in deeper medulla position and subsequent M6 layer targeting.

to the M6 and M3 layer respectively (*Ting and Lee, 2007*; *Özel et al., 2015*). To determine how the initial growth cone positioning influences subsequent steps of synaptic layer selection we followed the development of R8 growth cones which have been displaced to deeper R7 position. Wild type R8 growth cones are positioned at the superficial medulla at 24 hr APF (*Figure 6A,A′*) and later target to layer M3 in the adult (*Figure 6B,B′*). A pulse of Sequoia expression until 24 hr APF and subsequent repression using Gal80$^{ts}$ leads to an initial shift of all R8 growth cones to the deeper R7 position (*Figure 6C,C′*). Interestingly, in the subsequent steps of synaptic layer targeting, these R8 axons, together with R7 axons, terminate in the layer M6 even without further Sequoia expression (*Figure 6D,D′*). Additionally, an early Sequoia pulse until 6 hr APF leads to a shift of 8–10 posterior R8 growth cones towards the deeper R7 medulla position (*Figure 6E,E′*). Following this pattern during the subsequent steps of medulla development reveals that R8 axons recognize their final target layer exactly according to their initial growth cone position (*Figure 6F,F′*), with 10 most posterior R8 axons targeting to the layer M6 (*Figure 6F′* arrowhead) and the remaining R8 axons terminating in the layer M3 (*Figure 6F′* arrow). We analysed if this R8 mis-targeting to layer M6 results from Sequoia-induced changes in the expression of Capricious and the known R8 guidance receptor Frazzled (*Pecot et al., 2014*; *Shinza-Kameda et al., 2006*; *Timofeev et al., 2012*). However, no difference in Capricious and Frazzled expression can be detected in R8 cells with elevated Sequoia levels (*Figure 6G-J′*), which is in line with earlier data showing that Sequoia does not influence cell-type specific differentiation programs of R7 and R8 (*Petrovic and Hummel, 2008*).

Interestingly, the main R7 synaptic target cell, Dm8, is already in close proximity of R7 growth cones immediately following the R7/R8 segregation and positioning of R7 growth cones in the deeper medulla (*Figure 6—figure supplement 1A–A′*), *Ting et al., 2014*. Therefore the Dm8 neurites and mis-positioned R8 growth cones are also in close proximity raising the possibility that these two cell types can directly interact with each other (*Figure 6—figure supplement 1B–B′*). To test if the ectopic R8 axons form synapses with Dm8 neurites in layer M6, we made use of the recently modified GFP reconstitution method (syb-GRASP; *Karuppudurai et al., 2014*). In this method, the GFP1-10 fragment is fused to the C-terminus of *Drosophila* n-synaptobrevin (n-syb) producing n-syb::spGFP1-10 chimera, resulting in reconstitution with GFP11 only after vesicle fusion (*Macpherson et al., 2015*) and a preferential labelling of 'active-synapses' rather than neuronal contacts made at any time during development. In wild type (*Figure 6K–K″*), the expression of pre-synaptic syb::spGFP1-10 in R7/R8 (*LGMR-Gal4*) and post-synaptic *LexAop-spGFP11::CD4*, in Dm8 cells (*OrtC1-3 LexA DBD, OrtC2B dVP16AD; Karuppudurai et al., 2014*; See Materials and methods for details), results in GRASP-positive R7->Dm8 connections in layer M6 (*Figure 6K″* arrows) and no GRASP-positive R8->Dm8 connections can be detected in layer M3 (*Figure 6K″* arrowheads). When R7 growth cones are retained at the R8 temporary layer via *UAS-Capricious$^{ID}$* expression using *sev-Gal4* (*Shinza-Kameda et al., 2006*), GRASP signal, demonstrating R7->Dm8 contacts, can be observed only in columns where 'escaper' R7 axons target to layer M6 (*Figure 6L″* arrows), but not in columns with mis-targeted R7 axons (*Figure 6L″* arrowhead). Surprisingly, GRASP-positive R8->Dm8, similar to R7->Dm8, connections can be detected in medulla layer M6 following the early shift of R8 growth cones to the deeper medulla position (*Figure 6M–M″*), indicating that both R7 and R8 can recognize Dm8 processes and form synaptic contacts.

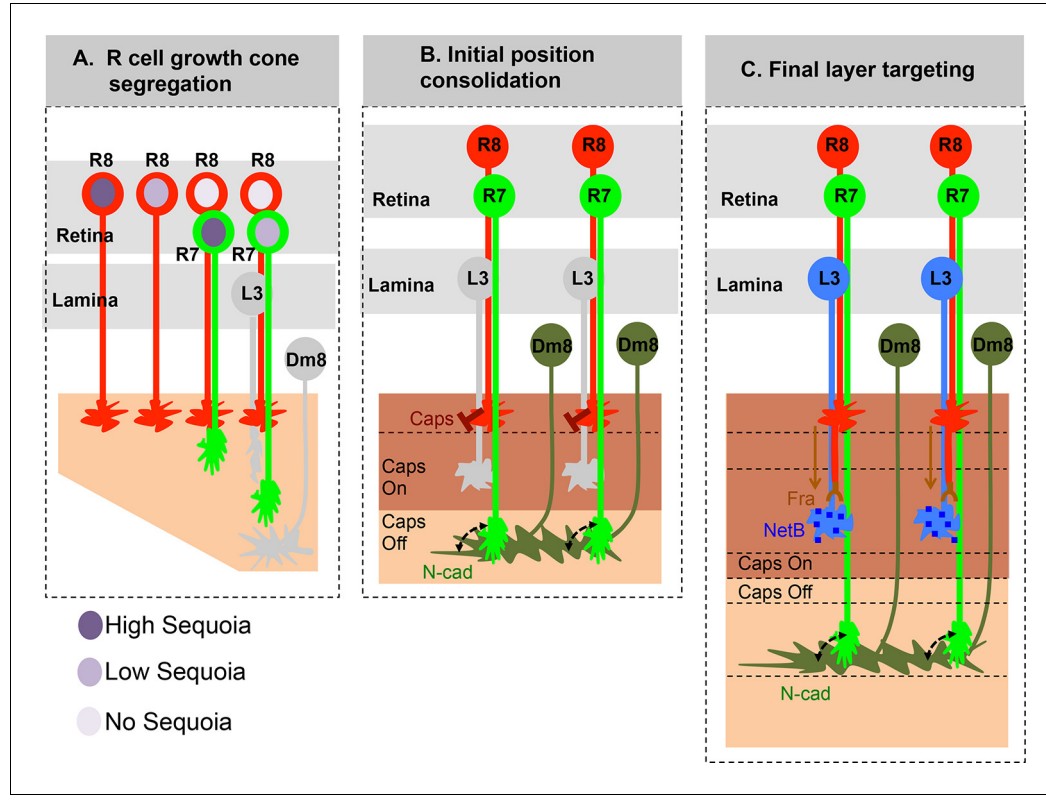

**Figure 7.** Role of early growth cone patterning in synaptic layer selection. (**A**) R cell axons arrive asynchronously in the developing medulla field in a pattern that reflects their specification in the eye disc. The growth cones of the same R cell type topographically segregate within the temporary layer. This segregation within a layer is mediated by low inter-ommatidial differences in the Sequoia levels (illustrated for R8 cells). The segregation of growth cones of different cell type (R7/R8) between distinct temporary layers occurs as a result high intra-ommatidial differences in Sequoia levels. The differences in Sequoia levels are an outcome of temporal sequence of photoreceptor specification in the eye disc (**Tomlinson and Ready, 1987**). Therefore, birth-order dependent differential positioning generates this early patterning of afferent growth cones in the medulla. (**B**) Once the initial patterning is achieved, the cell adhesion molecules consolidate the growth cones in their distinct temporary layers. In case of R8 growth cones, Capricious mediates afferent-target interactions and thereby stabilizes R8 growth cones in the superficial medulla position. The R7 growth cones, segregated into deeper medulla position most likely interact with neurites of Dm8 cells, the primary post-synaptic target of R7 cells via N-Cadherin (**Özel et al., 2015**). This sequential process continues until all R cell axons arrive in the developing medulla field, are segregated into and consolidated in distinct temporary layers. (**C**) Following the Capricious mediated consolidation of R8 growth cones in the superficial medulla and possible R7->Dm8 interaction in the deeper medulla position, axons of both cell types synchronously extend towards their final target layers. The guidance of R8 axons to M3 layer is mediated by localised NetB signal from L3 neurites (**Timofeev et al., 2012**; **Pecot et al., 2014**). On the contrary, the R7 axons are suggested to passively dislocate and reach layer M6 via their interactions with Dm8 neurites that are gradually pushed deeper in the medulla as a result of growth of the medulla field (**Özel et al., 2015**). Furthermore, the extension of R7 axons to reach layer M6, following their initial positioning, requires N-cadherin function (**Özel et al., 2015**; **Ting et al., 2005**). Therefore, M6 targeting of R7 axons seems to be a result of pre- and post-synaptic neuron interactions mediated by general synaptogenetic molecules rather than cell type specific factors. Thus, cellular proximity determines M6 layer targeting and establishment of synaptic contacts.

## Discussion

Here we demonstrate that the early, birth order dependent, segregation of R cell growth cones determines later synaptic layer identity in the *Drosophila* visual system. Small inter-ommatidial differences in Sequoia levels organize R cell growth cones within a layer whereas large intra-ommatidial differences segregate growth cones between layers. Changes in the positioning of growth cones directly correlate with changes in synaptic layer selection without affecting the expression of known

cell-type specific targeting molecules. These results highlight the importance of initial afferent growth cone positioning for visual map formation prior to synaptic partner recognition (*Figure 7*).

An early shift in the R8 growth cones to R7 position, induced by a short pulse of Sequoia expression, allows them to recognize the R7 target cell Dm8 as synaptic partner later during the development. Similarly, if R7 growth cones fail to segregate from R8 growth cones, they terminate together with R8 axons in the M3 synaptic layer independent of their intrinsic differentiation and targeting program. This extension of Frazzled-negative R7 axons (*Timofeev et al., 2012*) towards layer M3 could be explained by the default setting of R7 axons to tightly fasciculate and follow the R8 pioneer axons towards the target region (*Fischbach and Heisinger, 2008*; *Meinertzhagen and Hanson, 1993*).

R cell growth cone segregation can be controlled by axon-target interactions or axon-axon interactions or both. Although there is no experimental demonstration for direct R7-R8 afferent interactions, the following set of data indicate that such interactions occur during development. Support for direct interaction between R7-R8 afferents comes from *Maurel-Zaffran et al., 2001*, where the expression of LAR as membrane tethered ligand in R8 cells alone (using an R8-specific Gal4 driver line) could induce a response from R7 axons, indicating a direct signalling between R8-R7 axons. Further, induced Capricious expression in R8 and R7 cells, in Capricious null background (therefore resulting in a target region without any Capricious expression) is sufficient to mis-target R7 axons to layer M3, again indicating a direct R7-R8 afferent interaction (*Berger-Müller et al., 2013*). Here we show that final target layer of *sequoia* mutant R7 axons depends on the targeting of R8 axons, further suggesting that mis-targeting of sequoia mutant R7 axons to ectopic synaptic layer is the consequence of segregation defect rather than a change in target layer recognition.

Interactions among afferent axons have been implicated in the assembly of visual and olfactory circuits in vertebrates as well as invertebrates (*Brown et al., 2000*; *Clandinin and Zipursky, 2002*; *Ebrahimi and Chess, 2000*; *Feinstein and Mombaerts, 2004*; *Komiyama et al., 2007*; *Petrovic and Schmucker, 2015*; *Sweeney et al., 2007*). It has recently been shown that Eph-Ephrin signalling mediates local sorting of RGC axons in mammalian visual system (*Brown et al., 2000*; *Suetterlin and Drescher, 2014*). Notch signalling was demonstrated to play a role in spacing of DCN cluster neuron axons via neighbour axon interactions (*Langen et al., 2013*). But, whether these afferent interactions influence synaptic partner recognition is not known.

Here we show that in the *Drosophila* visual system, relative levels of Sequoia determine the segregation of afferent R7/R8 growth cones within or between layers. By creating Seq$^{high}$-Seq$^{low}$ R cell combinations using Sequoia gain-of-function R7 mosaics we observed that difference in Sequoia levels among neighbouring cells could induce growth cone segregation. The endogenous differences in Sequoia levels most likely arise as a result of the temporal sequence of R cell specification, suggesting a self-patterning mechanism in early visual circuit assembly (*Hassan and Hiesinger, 2015*; *Roignant and Treisman, 2009*, *Tomlinson and Ready, 1987*). How the relative differences in Sequoia levels in the nuclei of R cells translate into growth cone segregation remains elusive. We have tested candidate signalling pathways including Semaphorin/Plexin (*Cafferty et al., 2006*; *Hsieh et al., 2014*; *Pecot et al., 2013*; *Yu et al., 2010*), TGF-beta ligand Activin and its receptor Baboon (*Ting et al., 2007*) and Notch (*Langen et al, 2013*) but did not find evidence for a critical role in initial growth cone segregation (data not shown). This suggests a so far unknown molecular mechanism in which the growth state of an axon is directly coupled to differential growth cone adhesion. As we could demonstrate a cell-autonomous function of Sequoia in R8 for columnar segregation as well as in R7 for layer segregation, we envision a mechanistic model related to the concept of cell competition, in which strong cell-cell interactions induce cell-autonomous responses (*Rhiner et al., 2010*).

The initial segregation of afferent growth cones into distinct positions is then consolidated by expression of Capricious in R8 axons in the same posterior-to-anterior pattern in which they arrive in the medulla. We speculate that Capricious mediated growth cone consolidation serves two purposes: 1. It removes the temporal difference in the arrival of R8 axons and 2. It maintains R8 axons in the position where they are responsive to subsequent NetrinB signal provided by L3 neurites (*Pecot et al., 2014*; *Timofeev et al., 2012*). This is supported by two different sets of results: First, as we show in this study, the displacement of R8 growth cones to deeper medulla position leads to their mis-targeting to layer M6 in spite of normal Frazzled expression in these R8 cells. Second, the ectopic expression of Frazzled in R7 cells cannot re-direct them to M3 layer in response to localized NetrinB signal present in the superficial position but R8 axons can be re-directed to a different layer

(M1/M2) by ectopic expression of localized NetrinB in a position deeper to the superficial R8 medulla position (*Nern et al., 2008*; *Pecot et al., 2013*; *Timofeev et al., 2012*). Taken together, these observations suggest that M3 layer targeting via L3-mediated NetrinB signalling requires R8 axons to be positioned superficially in the medulla further underscoring the importance of R8 growth cone consolidation in this position (*Figure 7*).

Previous studies have identified several molecules necessary for M6 targeting of R7 axons (*Clandinin et al., 2001*; *Hofmeyer and Treisman, 2009*; *Hofmeyer et al., 2006*; *Maurel-Zaffran et al., 2001*; *Ting et al., 2005*; *Tong et al., 2011*), including Liprin-alpha/beta/gamma, PTP69D and D-Lar. The loss of these molecules specifically affects the stabilization of R7 growth cones during the second step of targeting. Additionally, these molecules along with N-Cadherin have been shown to be critical for establishment of synaptic contacts between pre and post-synaptic neurons (*Arikkath and Reichardt, 2008*; *Astigarraga et al., 2010b*; *Garrity et al., 1999*; *Hummel and Zipursky, 2004*; *Lee and Godenschwege, 2015*; *Nagaoka et al., 2014*; *Prakash et al., 2010*; *Reines et al., 2012*). We confirm previous observations, that R7 growth cones are in close proximity with their primary post-synaptic target neurons, Dm8, at the end of growth cone segregation (*Figure 6—figure supplement 1A–B'*, *Ting et al., 2014*). This raises the possibility that targeting of R7 axons to M6 layer, later in the development, could be the direct result of R7->Dm8 contacts mediated by N-Cadherin. Recently it was shown that N-Cadherin function is necessary for stabilizing R7 growth cones in the deeper medulla position but not for targeting and subsequent extension of R7 axons to M6 layer seems to be a result of passive dislocation (*Özel et al., 2015*). Additional support for the role of N-Cadherin in the formation and maintenance of R7->Dm8 contacts, following their initial segregation from R8 growth cones, comes from our observation that early expression of Sequoia in *CadN* mutant R7 cells under weak *elav-Gal4* driver can rescue the mis-targeting of R7 growth cones in the superficial medulla position along with R8 growth cones at 24 hr APF (*Figure 6—figure supplement 2A,A',C,C'*), but fails to rescue the later mis-targeting to layer M3 eventually resulting in a mis-targeting phenotype identical to *CadN* mutant R7 axons (*Figure 6—figure supplement 2B,B',D,D'*). Interestingly, the R8 growth cones initially mis-positioned in the deeper medulla eventually mis-target to layer M6 and form synaptic contacts with Dm8. In addition, these R8 cells, with axons mis-targeted to layer M6, do not show changes in any of their known cell-type specific molecules including early specifier of cell identity (Senseless), guidance receptors (Frazzled, Capricious) and sensory receptors (Rh6). In addition, no expression of R7 specific molecules (Prospero, R3, Rh4) can be detected. Thus, the R8 cells interact with Dm8 neurons most likely via ubiquitously expressed molecules such as N-Cadherin expressed in both, R7 as well as R8, cells. This is supported by our observation that N-Cadherin is required for stabilization of R8 axons at the layer M6 (*Figure 6—figure supplement 2E–F'*).

We observed that R8 axons form functional synapses with Dm8, a known R7 target neuron, in the layer M6. This raises the fundamental question of how synaptic layer selection influences synaptic partner recognition. The cellular complexity of potential post-synaptic target layer encountered by ingrowing R cell axons has not been fully determined, leaving room for selective recognition for synaptogenesis within a layer. In fact, it has been shown that within M6, R7 axons form synapses with Dm8 but not with Tm5c (*Karuppudurai et al., 2014*) which also arborize the M6 layer. In addition, we have identified various medulla columnar neurons within M6 that are not contacted by R7 axons (unpublished results). Similarly in layer M3 R8 and L3 select distinct post-synaptic partners (*Takemura et al., 2013*).

The types of neurons present in the medulla at the time of R8 and R7 axon innervation have not been fully identified. Based on the data from *Hasegawa et al., 2011*; *Li et al., 2013*; *Suzuki et al., 2013*, the medulla neurons are generated in temporal fashion (reviewed in *Sato et al., 2013*; *Suzuki and Sato, 2014*) and therefore they likely innervate the medulla at different time points. Experiments presented here support a developmental scenario in which the medulla context for arriving R cell axons reduces the complexity of synaptic partner selection. For example R8 and L3 have different arrival times at M3 (*Nern et al., 2005*; *Pecot et al., 2013*; *2014*), thereby would encounter a different local environment of potential post-synaptic partners competent for synaptogenesis. It is plausible that some form of temporal co-ordination of afferent axons and their post-synaptic partner cell neurites would actually simplify the synaptic partner matching. The concept of temporal identity would argue that R7 and R8 axons arriving at the same medulla position approximately the same time, as shown in the Sequoia gain-of-function background, will pick the same

synaptic partners exemplified by Dm8. Support for such proximity-based axon-target interaction for synaptogenesis comes from earlier analysis of ectopic axons in *Drosophila* as well as Zebrafish (*Edwards and Meinertzhagen, 2009*; *Berger-Müller et al., 2013*; *Pujol-Martí et al., 2014*).

From an evolutionary perspective, such proximity-induced synapse formation has several advantages over mechanisms that require regulation and expression of distinct sets of cell recognition molecules. Considering R7 as the most recently added cell to the precursor ommatidium (*Mavromatakis and Tomlinson, 2012*): During development, R7 is recruited using mechanisms similar to R8 and therefore possesses default R8 specification program. However, this default R8 program is suppressed to facilitate R7 specification (*Cook et al., 2003*; *Morey et al., 2008*). Thus, a temporally separated, novel R7 cell is generated with basic neuronal differentiation similar to that of an R8 cell (*Brennan and Moses, 2000*; *Friedrich et al., 2011*; *Roignant and Treisman, 2009*). Interestingly the temporal difference in the R8/R7 differentiation is then translated into Sequoia mediated layer segregation of their growth cones (*Petrovic and Hummel, 2008*), with Sequoia expression being part of common differentiation program. Thus, the evolutionary recent R7 cell seems to recognize its synaptic targets via pan neuronal molecules like N-Cadherin as part of the default neuronal differentiation program, instead of the invention of an additional recognition code.

## Materials and methods

### Fly stocks and Fly rearing

The flies were raised at 25°C unless otherwise mentioned.

The following flies were used in this study:

CantonS, FRT42B, FRT2A, Gal80/II, Gal80/III, FRT80, GMR-GFP, UAS-mCD8GFP, hs-FLP, ey3.5-FLP, GMR-FLP, tub-Gal80ts, elav-Gal4, sev-Gal4, LGMR-Gal4, m$\delta$0.5-Gal4, ro-tau-LacZ, Rh6-EGFP, PanR7-Gal4 were obtained from Bloomington *Drosophila* Stock Center. *sens-Gal4*/CyO and *sens-Gal4*/TM6 were gift from Bassem Hassan. *sequoia*[5] is a previously generated Sequoia loss-of-function allele (*Petrovic and Hummel, 2008*). UAS-Sequoia was obtained from Jay Brenman. *UAS-Capricious, UAS-Capricious*[ID](intracellular deletion), *Capricious*[C18fs] *FRT2A* and *caps-LacZ*[nls] flies were kindly provided by Akinao Nose. *GMR-gogo* was a generous gift from Takashi Suzuki. *PanR8-Gal4* was a gift from Claude Desplan. *UAS-Capricious*[RNAi] was obtained from VDRC (VDRC Transformant ID No. 27097). *PM181-Gal4* and *CadN*[405]*FRT40* (*Lee et al., 2001*) were used in Sequoia analysis, Early Dm8 labelling *OK371-VP16AD/CyO; ortC2-Gal4DBD/TM2*, adult Dm8 specific *OrtC1-3 LexA DBD, OrtC2B dVP16AD/CyO* (*Ting et al., 2014*) and *syn-GRASP constructs UAS-Syb::spGFP1-10 and LexAop spGFP11::CD4/TM2* (*Karuppudurai et al., 2014*) were used in syb-GRASP experiments.

### Antibodies used in this study

The primary antibodies used were: Rabbit anti-GFP (1:1000, Invitrogen, Carlsbad, California, USA), Mouse anti-GFP (1:100, Invitrogen), Chicken anti-GFP (1:1000 Abcam), Mouse 24B10 anti-Chaoptin (1:50, DSHB), Rat anti-CadN (1:20, DSHB), Rabbit anti-Sequoia (1:1000, *Brenman et al., 2001*), Mouse Anti-Elav (1:20 DSHB), Rabbit anti-LacZ (1:200, Invitrogen), Guinea pig anti-Senseless (1:1000) was kindly provided by Hugo Bellen, Guinea pig anti-Repo (1:200) and Rabbit anti-Frazzled (1:10) were kindly provided by Benjamin Altenhein.

Rabbit anti-Caps (1:50) antibody was generated for this study (See below).

The Secondary antibodies used were: Goat Anti-Rabbit Alexa-488 (1:500), Goat anti-Rabbit Alexa-568 (1:300), Goat anti-Mouse Alexa-488 (1:300), Goat anti-Mouse Alexa-560 (1:500), Goat anti-Mouse Alexa-647 (1:500), Goat anti-Rat Alexa-647 (1:300), Goat anti-Guinea pig Alexa-568 (1:500), Goat anti-Guinea pig Alexa-647 (1:300). All secondary antibodies were obtained from Invitrogen (Carlsbad, CA).

### Immunohistochemistry

Pupal and adult brains were dissected in PBS and fixed with 3.7% formaldehyde in PBS for 20 min. Fixed brains were blocked with 10% Goat Serum for one hour and then incubated with primary antibody in 10% Goat Serum (in 0.3% PBS-T) over night at 4°C. Following three times washing (15 min each), brains were incubated with secondary antibody diluted in 0.3% PBS-T overnight at 4°C. After three times washing (20 min each) brains were mounted in Vectashield (Vector Laboratory,

Burlingame, CA) anti-fade mounting medium for confocal microscopy. Images were obtained using a Leica TCS SP5II confocal microscope and processed with ImageJ and Adobe Photoshop CS5.1.

## Clonal analysis

MARCM clones were generated as previously described (*Lee and Luo, 1999*). Briefly, *ey3.5-* and *GMR-FLP* were used to generate eye and R7 specific clones, respectively. To generate R8 specific clones *FLP* was expressed under heat shock responsive promoter (*hs-FLP*). 2nd instar larvae were collected and subjected to heat shock at 37°C for 25 min (for large clones) or 5 min (single cell clones). Post heat shock, the larvae were incubated at 25°C, following a brief incubation at 18°C for half an hour, and brains were dissected at appropriate pupal stages.

## Generation of anti-Caps antibody

To generate an antibody specific against Capricious, we used the C-terminal peptide sequence AAGGYPYIAGNSRMIPVTEL as the antigen (*Shishido et al., 1998*). Using the standard services from GenScript (www.genscript.com), we generated a rabbit polyclonal antibody against the specified peptide. The specificity of the antibody was tested using immunostaining in the wing disc. The antibody was then used in the brain tissue after standardization for concentration.

## Fluorescence intensity measurement for Sequoia expression quantification

The normalized fluorescence intensity (NFI) was measured as described in *Komiyama et al. (2007)* using ImageJ. Briefly, the images were de-convoluted and fluorescence intensity of region of interest was normalized against the background of the image. The fluorescence intensity for each R8 cell in the same row was measured and normalized against the background. The ratio of average intensity for all R8 cells in the row and average intensity of same amount of area in the background was taken as the average Sequoia expression in R8 cells in that row. The average intensity of different R8 cell rows from multiple images was measured and plotted.

## Light induction of Synaptic GRASP

For visualization of synaptic contacts, syb-GRASP method was utilized as previously described (*Karuppudurai et al., 2014*). Briefly, the larvae and pupae were kept on a 12 hr light/dark cycle before eclosion. Following eclosion, the flies were raised in constant light (50lx (14W cm$^{-2}$) condition for 3–5 days. The brains were dissected in 4% PFA to avoid diffusion of the reconstituted GFP signal. Following 20 min incubation in 4% PFA, the brains were processed and imaged as previously described.

## Acknowledgements

We would like to thank all members of the Hummel lab for critical reading of the manuscript and helpful discussions. Special thanks to Bassem Hassan for exchanging many conceptual ideas. We further thank Jay Brenman and Milan Petrovic for providing critical reagents for this study. This work was supported by grants from the DFG and the Schram Foundation.

## Additional information

### Funding

| Funder | Grant reference number | Author |
|---|---|---|
| Deutsche Forschungsgemeinschaft | Hu992/1-2 | Thomas Hummel |
| Schram Foundation | BCA01 | Thomas Hummel |

The funders had no role in study design, data collection and interpretation, or the decision to submit the work for publication.

## Author contributions
AK, Conception and design, Acquisition of data, Analysis and interpretation of data, Drafting or revising the article; DE, Acquisition of data, Analysis and interpretation of data; C-HL, Conception and design, Contributed unpublished essential data or reagents; TH, Conception and design, Analysis and interpretation of data, Drafting or revising the article

## Author ORCIDs
Deniz Ertekin, http://orcid.org/0000-0001-9480-8622

## Additional files

### Supplementary files
• Supplementary file 1. List of genotypes used in this study. The table shows detailed genotypes used in each of the experiments shown in figures and arranged to depict genotypes analysed for each representative image in the figures.

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
