## [Decision Letter]

Thank you for submitting your work entitled "Birth Order Dependent Growth Cone Segregation Determines Synaptic Layer Identity in the *Drosophila* Visual System" for consideration by *eLife*. Your article has been favorably evaluated by K VijayRaghavan as Senior editor and three reviewers, one of whom is a member of our Board of Reviewing Editors.

The reviewers have discussed the reviews with one another and the Reviewing Editor has drafted this decision to help you prepare a revised submission.

Kulkarni et al. present an interesting study on layer-specific targeting through 'temporal coding' in the fly visual system. Specifically, they re-investigate the role of the transcription factor sequoia in determination of R7/8 target layer selection in medulla. The work builds on and significantly expands on previous work (Petrovic and Hummel, 2008). The authors show that photoreceptors can use seq levels as a direct molecular representation of their birth order and use this information to achieve layer and column specific targeting.

The strength of the paper lies in the novel and thorough analysis of the role and 'coding' of birth order of neurons in their targeting. This is especially attractive because it formulates a relatively simple developmental algorithm that controls several aspects of neural circuit assembly. The weakness, however, lies in the authors’ statement that the pre-patterning they describe controls 'synaptic partner matching'. In particular, neither the afferent-afferent interaction data nor the seq levels data are as strong as presented, leaving room for different downstream molecular explanation. Similarly, a simple layer-selection is not sufficient to pre-determine synaptic partners in a crowded, complicated environment. Subsequent patterning processes as well as molecular matchmaking may occur to ultimately specify synapses. Hence, while the core data on seq-dependent pre-patterning in layer formation remain a strong and important contribution, the authors should clarify and focus the paper on its core for a strong contribution to *eLife* and the field.

The notion that axon/axon interaction is important for layer-specificity is well established. It is most clearly demonstrated in a series of experiments published in two papers, Timofeev et al. (2012) and Pecot et al. (2014). Here the L3 growth cone in the M3 layer is essential for the targeting of R8. This is, at least, in part mediated by Netrin signaling; Netrin is expressed selectively in M3 and its receptor Frazzled is expressed and required in R8. The data presented in this paper does not in any way critically address the issue of whether R7/R8 interactions (whether mediated by seq or not; of course only indirectly as Seq is a transcription factor) play a role in promoting their segregation into different layers. This is one possible explanation of an interesting set of data but by no means the only one. At a minimum, demonstrating a role for growth cone/growth cone interaction requires generating one mutant growth cone with no defect itself leading to changes in the behavior of a neighboring wild type growth cone. This alone, however, does not provide strong support for a direct interaction; the medulla neuropile even at this early stage of development contains the processes of many different neuronal cell types. Thus, the interactions leading to appropriate segregation may be very indirect. The experiments presented in the paper are well done and interesting. But the interpretations are simply not straightforward given the complexity of the experimental design on one hand and the inherent complexity of the tissue. In short, the notion that R7/R8 interactions play an important role in their segregation is fascinating but the authors provide no compelling data in support of this idea.

The authors argue that these interactions are sufficient to determine synaptic specificity. That is, layer-specificity determines synaptic specificity. With the connectome in hand from the Janelia group there are many examples that this is simply not the case. For instance, R8 and L3 terminate in M3 but only L3 makes synaptic connections with Tm9. Each layer contains the processes of many different neurons and yet R7 and R8 within their respective layers only make synaptic connections with a discrete subset of them. There is simply no correlation between the area of contact and the probability of synaptic connectivity. The authors provide one example by showing that R8s misexpressing Seq, within a particular time frame during development, will mistarget to M6 and here, using a clever variation on GRASP they demonstrate synapses between R8 and Dm8, the normal target for R7. Based on this observation, and solely on this observation, they conclude that synaptic specificity is determined by the layer in which a terminal finds itself rather than through selective recognition between processes of different neurons within the same layer. The assumption here, of course, is that expression of Seq has no effect on R8 terminals other than re-positioning it to M6; that is the authors assume that an otherwise normal R8, that is mispositioned, is forming synapses with a target appropriate for this layer. It is simply not plausible that over expression of a transcription factor that is leading to a change in the position of the growth cone is also not changing its molecular nature. So while the results with sequoia mis-expression are consistent with the notion that layers are determinative for specificity, the data are merely a correlation and one resting on the assumption that mis-expression of Seq during a specific time period has not activated the expression of proteins in these mutant R8 cells for competence to form synapses with Dm8.

A key idea in the paper are tight interactions between R7/8 afferents. This is first stated in the second paragraph of the subsection “Sequoia mediates R cell growth cone segregation”, but the evidence is not obvious. Later in the manuscript, the authors propose that R7 axons get stuck in M1 in flies that are both seq mutant and *gogo* overexpressing is their attachment to R8s. However, *gogo* is overexpressed with the GMR promoter in these experiments. This means *gogo* may affect R7s independently from R8s, resulting in their failure to extend to the M3 later. The authors further state that (Discussion, second paragraph) seq mutant R7 axons extend to M3 later because of their association with R8 axons (also reference need for statement that R7 cells are Frazzled-negative). However, *CadN* mutant R7 axons that have previously retracted are capable of re-extension in the medulla seemingly independent of what R8s are doing and clearly at the wrong time (days later!) (Ozel et al. 2015), which at least raises the possibility that R7s are capable of directing themselves to M3 if they fail to stabilize in M6. A simple experiment here would be to overexpress *gogo* specifically in seq mutant R8s and observe the behavior of seq mutant R7 cells.

The authors use the term 'temporal layer' equally for R7 and R8. However, R8 indeed actively extends from an initially targeted temporary layer, whereas R7 remains stabilized (seemingly in stable interaction with Dm8) throughout medulla expansion (Ting et al., 2014; Ozel et al., 2015), while intercalation of other cells expand the medulla. The wording 'temporal layer' does not apply to R7, since it remains stabilized throughout development and doesn't move anywhere. The authors should come up with a different term for the 'temporary' state of R7 and R8 in this system. One interpretation of the data is that R7 axons as well as Seq overexpressing R8 axons target directly to the future M6 layer, which significantly simplifies the developmental problem at hand. The apparent movement or 'elongation' happens only because the entire M6 layer is pushed further down by the newly innervating lamina and medulla cells forming the other layers. The authors should replace that with "R7 target layer", "the future M6" or even simply M6.

The presentation of Seq^low^/Seq^high^ experiments is a bit confusing. The aims, methods and conclusions might be made a bit clearer. The model based on "relative levels" here is not obvious. Relativity implies that these axons have a way of communicating their Seq expression level to each other, but I am not sure there is evidence for this? If Seq levels are decreased or increased equally in all cells involved, there should be no defects as the "relative" levels do not change. To support the hypothesis of 'relative level comparison' one might present an experiment where Seq expression level changes in a subset of cells cause a defect but the same amount of change in all cells does not. If such an experiment is not feasible, the arguments and discussions dealing with comparisons of relative levels should be re-evaluated.

It is surprising that the authors observed Seq expression reduced the early-stage R7 retractions in *CadN* mutants since they had previously proposed sequoia action through modification of *CadN* levels (Petrovic and Hummel, 2008). Kulkarni et al. now provide a picture of a single R7 cell with normal location in Figure 6—figure supplement 6B. However, the ratio of retracted R7 axons at P24 is actually quite low (very likely to see a normal R7 cell no matter what). A quantitative analysis of mistargeted R7 axons (n=>50) would be necessary to argue that Seq overexpression somehow alleviates the R7 *CadN* phenotype.

---

## [Author Response]

Kulkarni et al. present an interesting study on layer-specific targeting through 'temporal coding' in the fly visual system. Specifically, they re-investigate the role of the transcription factor sequoia in determination of R7/8 target layer selection in medulla. The work builds on and significantly expands on previous work (Petrovic and Hummel, 2008). The authors show that photoreceptors can use seq levels as a direct molecular representation of their birth order and use this information to achieve layer and column specific targeting.

*The strength of the paper lies in the novel and thorough analysis of the role and 'coding' of birth order of neurons in their targeting. This is especially attractive because it formulates a relatively simple developmental algorithm that controls several aspects of neural circuit assembly. The weakness, however, lies in the author's statement that the pre-patterning they describe controls 'synaptic partner matching'. In particular, neither the afferent-afferent interaction data nor the seq levels data are as strong as presented, leaving room for different downstream molecular explanation. Similarly, a simple layer-selection is not sufficient to pre-determine synaptic partners in a crowded, complicated environment. Subsequent patterning processes as well as molecular matchmaking may occur to ultimately specify synapses. Hence, while the core data on seq-dependent pre-patterning in layer formation remain a strong and important contribution, the authors should clarify and focus the paper on its core for a strong contribution to eLife and the field. The notion that axon/axon interaction is important for layer-specificity is well established. It is most clearly demonstrated in a series of experiments published in two papers, Timofeev et al. (2012) and Pecot et al. (2014). Here the L3 growth cone in the M3 layer is essential for the targeting of R8. This is, at least, in part mediated by Netrin signaling; Netrin is expressed selectively in M3 and its receptor Frazzled is expressed and required in R8. The data presented in this paper does not in any way critically address the issue of whether R7/R8 interactions (whether mediated by seq or not; of course only indirectly as Seq is a transcription factor) play a role in promoting their segregation into different layers.*

The developmental step of growth cone segregation between R7 and R8 we describe in the manuscript can be controlled by axon-target interaction, axon-axon interaction or both. Although there is no experimental demonstration of direct R7-R8 afferent interactions, the following set of data indicate that such interactions occur during development. 1) It is well established that R7 axons tightly fasciculate and follow R8 pioneer axons from the eye disc into the target region (Meinertzhagen and Hanson, 1993). 2) Further support for direct interaction between R7-R8 afferents comes from Maurel-Zaffran et al., 2001, where the expression of LAR as a membrane tethered ligand in R8 cells alone (using an R8-specific *Gal4* driver line) can induce a response from R7 axons, indicating a direct signalling between R8-R7 axons. 3) In addition, induced Capricious expression in R8 and R7 cells, in a Capricious null background (therefore resulting in a target region without any Caps expression) is sufficient to mis-target R7 axons to layer M3, again indicating a direct R7-R8 afferent interaction (Berger-Müller et al., 2013). In addition, in the manuscript we show that the final target layer of sequoia mutant R7 axons depends on the targeting of R8 axons, suggesting that mis-targeting of sequoia mutant R7 axons to ectopic synaptic layer is the consequence of segregation defects rather than a change in target layer recognition.

*This is one possible explanation of an interesting set of data but by no means the only one. At a minimum, demonstrating a role for growth cone/growth cone interaction requires generating one mutant growth cone with no defect itself leading to changes in the behavior of a neighboring wild type growth cone.*

This would be the case for a non-cell autonomous function, e.g. in the case of a Seq-induced repulsive signal. In a scenario in which Seq is critical in the responding cell, e.g. by modulating the amount or downstream signalling of a receptor, one would expect a cell-autonomous function. As we could demonstrate a cell-autonomous function of Seq in R8 for columnar segregation as well as in R7 for layer segregation we envision a mechanistic model related to the concept of cell competition, in which strong cell-cell interactions resulting in cell-autonomous responses (discussed in the manuscript).

*This alone, however, does not provide strong support for a direct interaction; the medulla neuropile even at this early stage of development contains the processes of many different neuronal cell types.*

The types of neurons present in the medulla at the time of R8 and R7 axon innervation remains unknown. Based on data from Li et al., 2013 and Suzuki T et al., 2013, the medulla neurons are generated in temporal fashion and therefore they likely innervate the medulla at different time points (Morante J and Desplan C, 2008, Morante J et al., 2011). We expect that some form of temporal coordination of afferent axons and their post-synaptic partner cell neurites would actually simplify the synaptic partner matching.

*Thus, the interactions leading to appropriate segregation may be very indirect. The experiments presented in the paper are well done and interesting. But the interpretations are simply not straightforward given the complexity of the experimental design on one hand and the inherent complexity of the tissue. In short, the notion that R7/R8 interactions play an important role in their segregation is fascinating but the authors provide no compelling data in support of this idea.*

We fully agree that the manuscript is not providing any data for direct R7/8 afferent interaction and we are pointing this out in the revised version (e.g. replacing “cell-cell interaction” by “growth cone segregation”). The indirect evidence supporting a role of R7/8 interactions are now discussed in more detail.

*The authors argue that these interactions are sufficient to determine synaptic specificity. That is, layer-specificity determines synaptic specificity. With the connectome in hand from the Janelia group there are many examples that this is simply not the case. For instance, R8 and L3 terminate in M3 but only L3 makes synaptic connections with Tm9. Each layer contains the processes of many different neurons and yet R7 and R8 within their respective layers only make synaptic connections with a discrete subset of them. There is simply no correlation between the area of contact and the probability of synaptic connectivity.*

We fully agree with the criticism and thank the reviewers for raising this issue. We did not intend to state that initial layer identity specifies synaptic partner matching in subsequent steps of development. The cellular complexity of potential postsynaptic target cells for R cell axons has not been fully determined, leaving room for selective mechanisms for synaptogenesis within a layer. In fact, it has been shown that within M6, R7axons form synapses with DM8 but not with Tm5c, which also arborizes in the M6 layer. In addition, we have identified various medulla columnar neurons within M6, which are not contacted by R7 axons (unpublished results). Similar for M3, in which R8 and L3 select distinct post-synaptic partner.

Our main conclusion from the experiments presented in the manuscript is that the local and temporal context of the early medulla target field reduces the complexity of synaptic partner selection for arriving R cell axons. For example R8 and L3 have different arrival times at M3 thereby encounter a different local environment of potential postsynaptic partners competent for synaptogenesis. On the other hand, the concept of temporal identity would argue that R7 and R8 arriving at about the same time (in the deeper medulla position in Sequoia gain of function studies) would pick the same partners, exemplified by Dm8.

*The authors provide one example by showing that R8s misexpressing Seq, within a particular time frame during development, will mistarget to M6 and here, using a clever variation on GRASP they demonstrate synapses between R8 and Dm8, the normal target for R7. Based on this observation, and solely on this observation, they conclude that synaptic specificity is determined by the layer in which a terminal finds itself rather than through selective recognition between processes of different neurons within the same layer.*

With the identification of ectopic R8-Dm8 synapses in M6 we did not intend to conclude that layer identity is equivalent to synaptic identity but rather as a proof of principle that ectopic growth cones are capable of forming functional synapses. This is in the line with previous observations that misplaced R cell axons can trigger synapse formation in ectopic regions (Edwards T and Meinertzhagen I, 2009). Actually with the simplified version of R7-Dm8 interaction at an early, and most likely less complex cellular environment compared to the mature M6 layer, we could envision a similar mechanism works for establishing ectopic R8-Dm8 contacts (most likely Ncad mediated as Ncad is necessary for M6 targeting of R8, Figure 6—figure supplement 2).

We thank the reviewers for raising this issue and realised, going through the manuscript, that we did not make a clear distinction between synaptic layer and synaptic partner choice. We now emphasize in the manuscript that a change in the synaptic layer choice and formation of synapses within ectopic layer are not synonymous to a complete switch in the synaptic partner recognition. Proposing a role for differential growth cone positioning, we have argued that temporal coding of afferent-target interactions would reduce the developmental complexity of synaptic partner recognition within a complex environment.

*The assumption here, of course, is that expression of Seq has no effect on R8 terminals other than re-positioning it to M6; that is the authors assume that an otherwise normal R8, that is mispositioned, is forming synapses with a target appropriate for this layer. It is simply not plausible that over expression of a transcription factor that is leading to a change in the position of the growth cone is also not changing its molecular nature. So while the results with sequoia mis-expression are consistent with the notion that layers are determinative for specificity, the data are merely a correlation and one resting on the assumption that mis-expression of Seq during a specific time period has not activated the expression of proteins in these mutant R8 cells for competence to form synapses with Dm8.*

Although we fully accept the possibility of changes in the expression of unknown molecules following prolonged Seq expression, so for there is no indication for a change in cellular identity of R cells. We have tested many molecular markers of R8 cell differentiation including expression of Senseless, Frazzled, Capricious, Rhodopsin6, which are not affected by Sequoia loss of function or gain of function. Similarly, R7 differentiation markers like Prospero and Rhodopsins. Thus, for both R7 and R8 early differentiation markers as well as late cell type markers do not change upon changes in Sequoia expression, either loss or gain. Together with the fact the all R cells show the same pattern of Sequoia expression, just in a different temporal sequence, we think it is less likely that the observed changes in synaptic connections are due to a specific switch in cellular identity but rather result from changes in the initial growth cone patterning. We have discussed this issue in the manuscript.

*A key idea in the paper are tight interactions between R7/8 afferents. This is first stated in the second paragraph of the subsection “Sequoia mediates R cell growth cone segregation”, but the evidence is not obvious. Later in the manuscript, the authors propose that R7 axons get stuck in M1 in flies that are both seq mutant and gogo overexpressing is their attachment to R8s. However, gogo is overexpressed with the GMR promoter in these experiments. This means gogo may affect R7s independently from R8s, resulting in their failure to extend to the M3 later.*

The issue of direct afferent-afferent interaction has been addressed above. Regarding the *gogo* experiment mentioned here, *gogo* expression in R7 cells alone does not induce any mis-targeting (Berger-Muller et al., 2013, Hakeda-Suzuki et al., 2011). Moreover, we confirmed that *GMR-gogo* which expresses *gogo* in both R7 and R8 cells does not does not lead to mis-targeting of wild type R7 axons as shown in Figure 2’, although R8 axons are restricted to layer M1. This indicates that *gogo* expression alone does not cause any defects in R7 targeting.

We expect that Sequoia mutation alone is not sufficient to redirect R7 cells to layer M1 as they do target to layer M3 when only R7 cells are mutated using GMR-FLP as shown in Figure 2’ as well as in Petrovic and Hummel, 2008. To check the possibility of ectopic interaction between R8 and R7 cells when both are Sequoia mutant leading to M1 targeting of R7 axons, we wanted to restrict R8 axons to M1 layer without making them Sequoia mutant. For this we utilized GMR-*gogo* and analysed if R7 axons are dependent on R8 axons for their targeting once they initially fail to segregate. The R7 growth cones that fail to segregate from R8 growth cones as a result of loss of Sequoia in R7s, target to the same layer as the R8 axons whether to M3 without *gogo* expression or to M1 in presence of *gogo* expression.

Under no circumstance, observed so far, R7 axons can target to layer M3 in the absence of R8 axons in layer M3. In all cases analysed – Ncad mutant (Ting et al., 2005, Özel et al., 2015), Sequoia mutant (Petrovic et al., 2008), ectopic Caps expression (Shinza-Kameda et al., 2006, Berger-Müller, 2013) and *gogo*, Fmi co-expression (Hakeda-Suzuki, 2011, Berger Müller et al., 2013), R7 axons mis-target to layer M3 only in presence of R8 axons in layer M3. There is no condition known so far that leads to R7 axon mis-targeting layer M3 without R8 axons being present in layer M3. These observations point to the direct interactions between R8 and R7 axons.

*The authors further state that (Discussion, second paragraph) seq mutant R7 axons extend to M3 later because of their association with R8 axons (also reference need for statement that R7 cells are Frazzled-negative).*

We thank the reviewers for pointing this out and have mentioned this in the manuscript. Timofeev et al., 2012 show that in an ommatidium only R8 expresses Frazzled.

*However, CadN mutant R7 axons that have previously retracted are capable of re-extension in the medulla seemingly independent of what R8s are doing and clearly at the wrong time (days later!) (Ozel et al. 2015), which at least raises the possibility that R7s are capable of directing themselves to M3 if they fail to stabilize in M6.*

We thank the reviewers for mentioning this set of independent data. We looked very carefully at the presented live imaging results presented by Özel et al., 2015 and in our opinion, the described re-extension of R7 axons occurs between R8 position to M6 layer rather than M1 to M3 or M6. So, the eventual mistargeting of R7 axons to M3 is a result of retraction from M6 to M3 rather an extension from M1 to M3. As pointed out above, these data also support a model in which R7 axons, which fail to stabilize with the target cell, retract to their default interacting partner, which is R8.

*A simple experiment here would be to overexpress gogo specifically in seq mutant R8s and observe the behavior of seq mutant R7 cells.*

We have previously shown in Petrovic and Hummel, 2008 that when R8 and R7 cells, within the same column, are *sequoia* mutant they both mis-target to layer M1, even in absence of *gogo* over-expression in R8. In this background, it will be difficult to distinguish if M1 targeting of *sequoia* mutant R7 cells was due to their interaction/dependence on R8 cells or due to their own inability to target to layer M3.

*The authors use the term 'temporal layer' equally for R7 and R8. However, R8 indeed actively extends from an initially targeted temporary layer, whereas R7 remains stabilized (seemingly in stable interaction with Dm8) throughout medulla expansion (Ting et al., 2014; Ozel et al., 2015), while intercalation of other cells expand the medulla. The wording 'temporal layer' does not apply to R7, since it remains stabilized throughout development and doesn't move anywhere. The authors should come up with a different term for the 'temporary' state of R7 and R8 in this system.*

This is a very important point raised by the reviewers highlighting the different scenarios for R8 and R7 axons, the active extension of R8 and the more passive extension of R7. Although this requires different cellular mechanisms, both R7 and R8 start from transient positions that are different in their medulla location and, most likely, in their cellular composition and complexity. Therefore, the temporal position and final target layer are different for both, R8 and R7. We use the common term temporary layer positioning of R7/8 for the outcome of the initial R cell axon patterning and target layer for their final synaptic position in the mature medulla.

*One interpretation of the data is that R7 axons as well as Seq overexpressing R8 axons target directly to the future M6 layer, which significantly simplifies the developmental problem at hand. The apparent movement or 'elongation' happens only because the entire M6 layer is pushed further down by the newly innervating lamina and medulla cells forming the other layers. The authors should replace that with "R7 target layer", "the future M6" or even simply M6.* We did not state a direct R7-Dm8 interaction because this has not been demonstrated yet, either from previous studies (Ting et al., 2014, Özel et al., 2015) nor do we provide any experimental data for this. But we have emphasised that R7 most likely contacts post-synaptic partner Dm8 immediately following the growth cone segregation. Which would also occur for displaced R8 axons that eventually mistarget to M6, thereby explaining the specificity for R8-Dm8 connections.

*The presentation of Seq^low^/Seq^high^ experiments is a bit confusing. The aims, methods and conclusions might be made a bit clearer. The model based on "relative levels" here is not obvious. Relativity implies that these axons have a way of communicating their Seq expression level to each other, but I am not sure there is evidence for this? If Seq levels are decreased or increased equally in all cells involved, there should be no defects as the "relative" levels do not change.*

This is an important point and we are glad that the reviewers are bringing this up because we think this is a misunderstanding and not phrased appropriately by us in the first version of the manuscript. In all described Seq gain-of-function experiments we do not increase the Seq level equally but we erase level differences between neighbouring R cells by prolonged Seq expression thereby equalizing Seq levels.

*To support the hypothesis of 'relative level comparison' one might present an experiment where Seq expression level changes in a subset of cells cause a defect but the same amount of change in all cells does not. If such an experiment is not feasible, the arguments and discussions dealing with comparisons of relative levels should be re-evaluated.*

This is a very interesting suggestion, which in our opinion, has two different aspects:

The scenario in which ‘Seq expression level changes in a subset of cells cause a defect’ is described in the manuscript where we, in LOF and GOF mosaics approaches, analysed the behaviour of R cell with no-Seq or high-Seq compared to their direct neighbours: 1) No Seq leads to no segregation; 2) Equally high Sequoia also prevents segregation; 3) Different scenarios of neighbouring R cells with induced relative Seq differences (high-low) triggers growth cone segregation.

Regarding the second condition to induce ‘the same amount of change in all cells’, instead of the relative amount is unfortunately not feasible with all existing genetic tools because it would require an expression system which fully recapitulates the endogenous dynamics of Seq, which does not exist so far.

Taken together, abolishing the relative differences of Seq expression in LOF- and GOF- approaches leads to a similar non-segregation phenotype, whereas the induction of differences in mosaic GOF scenarios can trigger growth cone segregation for multiple R cell types (in medulla as well as in lamina). This is a strong indication that relative differences are critical to translate sequential axon arrival into growth cone segregation.

*It is surprising that the authors observed Seq expression reduced the early-stage R7 retractions in CadN mutants since they had previously proposed sequoia action through modification of CadN levels (Petrovic and Hummel, 2008).*

Yes, together with the previous observation that Ncad expression is not sufficient to rescue *sequoia* mutant R7 phenotype, this finding supports our model in which Seq-mediated growth cone segregation is preceding the N-cad mediated axon-target interaction.

Kulkarni et al. now provide a picture of a single R7 cell with normal location in Figure 6—figure supplement 6B. However, the ratio of retracted R7 axons at P24 is actually quite low (very likely to see a normal R7 cell no matter what). A quantitative analysis of mistargeted R7 axons (n=>50) would be necessary to argue that Seq overexpression somehow alleviates the R7 CadN phenotype.

Thank you, we missed to mention this. The quantification of the observed phenotypes is as follows: 1) Ncad mutant R7 growth cones without Sequoia- 71.42%, n=91 (65/91 show mis-targeting). 2) Ncad mutant R7 growth cones with Sequoia- 12.14%, n=107 (13/107 show mis-targeting). Quantified in the figure as the number of axons showing the rescue: 87.85% n=107 (94/107). This is a significant improvement in number of R7 growth cones positioned in the correct layer as compared to the N-Cadherin mutant axon number, which is approximately 30%.